# Interpretations of Domain Adaptations via Layer Variational Analysis

**Huan-Hsin Tseng, Hsin-Yi Lin, Kuo-Hsuan Hung, Yu Tsao**
Research Center for Information Technology Innovation, Academia Sinica, Taiwan
`{htseng, hylin, khhung, yu.tsao}@citi.sinica.edu.tw`

## Abstract

Transfer learning is known to perform efficiently in many applications empirically, yet limited literature reports the mechanism behind the scene. This study establishes both formal derivations and heuristic analysis to formulate the theory of transfer learning in deep learning. Our framework utilizing layer variational analysis proves that the success of transfer learning can be guaranteed with corresponding data conditions. Moreover, our theoretical calculation yields intuitive interpretations towards the knowledge transfer process. Subsequently, an alternative method for network-based transfer learning is derived. The method shows an increase in efficiency and accuracy for domain adaptation. It is particularly advantageous when new domain data is sufficiently sparse during adaptation. Numerical experiments over diverse tasks validated our theory and verified that our analytic expression achieved better performance in domain adaptation than the gradient descent method.

## 1 Introduction

Transfer learning is a technique applied to neural networks admitting rapid learning from one (source) domain to another domain, and it mimics human brains in terms of cognitive understanding. The concept of transfer learning has been considerably advantageous, and different frameworks have been formulated for various applications in different fields. For instance, it has been widely applied in image classification (Quattoni et al., 2008; Zhu et al., 2011; Hussain et al., 2018), object detection (Shin et al., 2016), and natural language processing (NLP) (Houlsby et al., 2019; Raffel et al., 2019). In addition to applications in computer vision and NLP, transferability is fundamentally and directly related to domain adaptation and adversarial learning (Luo et al., 2017; Cao et al., 2018; Ganin et al., 2016). Another majority field adopting transfer learning is domain adaptation, which investigates transition problems between two close domains (Kouw & Loog, 2018). A typical understanding is that transfer learning deals with a general problem where two domains can be rather distinct, allowing sample space and label space to differ. While domain adaptation is considered a subfield in transfer learning where the sample/label spaces are fixed with only the probability distributions allowed to be varied. Several studies have investigated the transferability of network features or representations through experimentation, and discussed their relation to network structures (Yosinski et al., 2014), features, and parameter spaces (Neyshabur et al., 2020; Gonthier et al., 2020).

In general, all methods that improve the predictive performance of a target domain, using knowledge of a source domain, are considered under the transfer learning category (Weiss et al., 2016; Tan et al., 2018). This work particularly focuses on network-based transfer-learning, which refers to a specific framework that reuses a pretrained network. This approach is often referred to as finetuning, which has been shown powerful and widely applied with deep-learning models (Ge & Yu, 2017; Guo et al., 2019). Even with abundant successes in applications, the understanding of the network-based transfer learning mechanism from a theoretical framework remains limited.

This paper presents a theoretical framework set out from aspects of functional variation analysis (Gelfand et al., 2000) to rigorously discuss the mechanism of transfer learning. Under the framework, error estimates can be computed to support the foundation of transfer-learning, and an interpretation is provided to connect the theoretical derivations with transfer learning mechanism.

Our contributions can be summarized as follows: we formalize transfer learning in a rigorous setting and variational analysis to build up a theoretical foundation for the empirical technique. A theorem is

proved through layer variational analysis that under certain data similarity conditions, a transferred net is guaranteed to transfer knowledge successfully. Moreover, a comprehensible interpretation is presented to understand the finetuning mechanism. Subsequently, the interpretation reveals that the reduction of non-trivial transfer learning loss can be represented by a linear regression form, which naturally leads to analytical (globally) optimal solutions. Experiments in domain adaptation were conducted and showed promising results, which validated our theoretical framework.

## 2   RELATED WORK

**Transfer Learning**   Transfer learning based on deep learning has achieved great success, and finetuning a pretrained model has been considered an influential approach for knowledge transfer (Guo et al., 2019; Girshick et al., 2014; Long et al., 2015). Due to its importance, many studies attempt to understand the transferability from a wide range of perspectives, such as its dependency on the base model (Kornblith et al., 2019), and the relation with features and parameters (Pan & Yang, 2009). Empirically, similarity could be a factor for successful knowledge transfer. The similarity between the pretrained and finetuned models was discussed in (Xuhong et al., 2018). Our work begins by setting up a mathematical framework with data similarity defined, which leads to a new formulation with intuitive interpretations for transferred models.

**Domain Adaptation**   Typically domain adaptations consider data distributions and deviations to search for mappings aligning domains. Early literature suggested that assuming data drawn from certain probabilities (Blitzer et al., 2006) can be used to model and compensate for the domain mismatch. Some studies then looked for theoretical arguments when a successful adaptation can be yielded. Particularly, (Redko et al., 2020) estimated learning bounds on various statistical conditions and yielded theoretical guarantees under classical learning settings. Inspired by deep learning, feature extractions (Wang & Deng, 2018) and efficient finetuning of networks (Patricia & Caputo, 2014; Donahue et al., 2014; Li et al., 2019) become popular techniques for achieving domain adaptation tasks. The finetuning of networks was investigated further to see what was being transferred and learned in (Wei et al., 2018). This will be close to our investigation on weights optimizations.

## 3   MATHEMATICAL FRAMEWORK AND ERROR ESTIMATES

### 3.1   FRAMEWORK FOR NETWORK-BASED TRANSFER LEARNING

To formally address error estimates, we first formulate the framework and notations for consistency.

**Definition 3.1** (Neural networks). *An $n$-layer neural network $f : \mathbb{R}^{\ell_0} \to \mathbb{R}^{\ell_n}$ is a function of the form:*

$$f = f_n \circ f_{n-1} \circ \cdots \circ f_1 \qquad (1)$$

*for each $j = 1, \ldots, n$, $f_j = \sigma_j \circ A_j : \mathbb{R}^{\ell_{j-1}} \to \mathbb{R}^{\ell_j}$ is a **layer** composed by an affine function $A_j(z) := W_j z + b_j$ and an activation function $\sigma_j : \mathbb{R}^{\ell_j} \to \mathbb{R}^{\ell_j}$, with $W_j \in L\left(\mathbb{R}^{\ell_{j-1}}, \mathbb{R}^{\ell_j}\right)$, $b_j \in \mathbb{R}^{\ell_j}$ and $L(K, V)$ the collection of all linear maps between two linear spaces $K \to V$.*

The concept of transfer learning is formulated as follows:

**Definition 3.2** ($K$-layer fixed transfer learning). *Given one (large and diverse) dataset $\mathcal{D} = \{(x_i, y_i) \in \mathcal{X} \times \mathcal{Y}\}$ and a corresponding $n$-layer network $f = f_n \circ f_{n-1} \circ \ldots \circ f_1 : \mathcal{X} \to \mathcal{Y}$ trained by $\mathcal{D}$ under loss $\mathcal{L}(f)$, the $k$-**layer fixed transfer learning** finds a new network $g : \mathcal{X} \to \mathcal{Y}$ of the form,*

$$g = g_n \circ g_{n-1} \circ \cdots \circ g_{k+1} \circ \underbrace{f_k \circ \cdots \circ f_1}_{\text{fixed}}, \qquad (2)$$

*under loss $\mathcal{L}(g)$ when a new and similar dataset $\widetilde{\mathcal{D}} = \{(\widetilde{x}_i, \widetilde{y}_i) \in \mathcal{X} \times \mathcal{Y}\}$ is given. The first $k$ layers of $f$ remain fixed and $g_j := \sigma_j \circ \widetilde{A}_j$ are new layers with affine functions $\widetilde{A}_j$ to be adjusted ($k < j < n$).*

The net $f$ trained by the original data $\mathcal{D}$ is called the **pretrained net** and $g$ trained by new data $\widetilde{\mathcal{D}}$ is called the **transferred net** or the **finetuned net**. Transfer learning has the empirical implication of

"transferring" knowledge from one domain to another by fixing the first few pretrained layers. One main goal of this study is to characterize the transferring process under layer finetuning.

We utilize the loss function to define the concept of a "*well-trained*" net, in order to comprehend the transfer learning mechanism. The losses for the pretrained net $f$ and the transferred net $g$ are computed on a general label space $\mathcal{Y}$ with norm $\|\cdot\|_{\mathcal{Y}}$:

$$\mathcal{L}(f) = \frac{1}{N} \sum_{i=1}^{N} \|f(x_i) - y_i\|_{\mathcal{Y}}^2, \quad \mathcal{L}(g) = \frac{1}{N} \sum_{i=1}^{N} \|g(\widetilde{x}_i) - \widetilde{y}_i\|_{\mathcal{Y}}^2. \tag{3}$$

**Definition 3.3** (Well-trained net). *Given a dataset $\mathfrak{D} = \{(x_i, y_i) \in \mathcal{X} \times \mathcal{Y}\}$, a network $h$ is called* **well-trained under data $\mathfrak{D}$ within error** $\varepsilon_{trained} > 0$ *if*

$$\mathcal{L}(h) = \sum_{i=1}^{N} \|h(x_i) - y_i\|_{\mathcal{Y}}^2 \leq \varepsilon_{trained}^2. \tag{4}$$

The definition then implies that a *well-pretrained* net $f$ within error $\varepsilon_{\text{pretrained}}$ is to satisfy

$$\mathcal{L}(f) \leq \varepsilon_{\text{pretrained}}^2. \tag{5}$$

In certain domain adaptation applications, discussions are in terms of probability distributions (Redko et al., 2020). Our framework is compatible with the usual probabilistic setting under the independent and identically distributed (i.i.d.) consideration, frequently applied in practice. In this case, Eq. (3) is equivalent to

$$\mathcal{L}(f) = \mathbb{E}_{(x,y) \sim \mathcal{D}} \, \ell^2(f(x), y), \quad \mathcal{L}(g) = \mathbb{E}_{(\widetilde{x}, \widetilde{y}) \sim \widetilde{\mathcal{D}}} \, \ell^2(g(\widetilde{x}), \widetilde{y}) \tag{6}$$

where $\ell$ is a norm (error) function.

One intuition for successful knowledge transfer is that the previous knowledge base and the new domain share some common (or similar) features. The resemblance of two datasets $\mathcal{D}$, $\widetilde{\mathcal{D}}$ under transfer learning in this work is defined by:

**Definition 3.4** (Data deviation). *The **data deviation** $\varepsilon_{data} \geq 0$ of two datasets $\mathcal{D}$ and $\widetilde{\mathcal{D}}$ is defined as,*

$$\max_i \left\{ \min_j \left\{ \|(\widetilde{x}_i, \widetilde{y}_i) - (x_j, y_j)\|_{\mathcal{X} \times \mathcal{Y}} \right\} \right\} \leq \varepsilon_{data}, \qquad (i \leq |\widetilde{\mathcal{D}}|, \; j \leq |\mathcal{D}|) \tag{7}$$

Note that the first minimization in Eq. (7) is recognized as performing *sample alignment* in the context of domain adaptation. In fact, for each $i \leq |\widetilde{\mathcal{D}}|$ there exists an index

$$j_i := \arg\min_j \left\{ \|(\widetilde{x}_i, \widetilde{y}_i) - (x_j, y_j)\|_{\mathcal{X} \times \mathcal{Y}} \right\} \tag{8}$$

such $j_i$ then corresponds to the *closet* sample in $\mathcal{D}$ to $i$. This step is usually referred to as **sample alignment**. **Domain alignment** is then to perform sample alignment over all samples of a given domain. Consequently, this definition aligns two datasets first and then considers the largest distance among all aligned pairs. Without loss of generality, in the rest discussions the sample index is rearranged by Eq. (8) with $j_i$ renamed as $i$ for convenience.

## 3.2 TRANSFER LOSS BOUND VIA LAYER VARIATIONS

Neural networks of the form Eq. (1) with Lipschitz activation functions are Lipschitz continuous. We utilize Lipschitz constants to control the network propagation and estimate the error of the last $r$-layer finetuned nets. The Lipschitz constant of a network $h$ is denoted by $C_h$ in this paper. Let a pretrained net $f$ be composed of two parts:

$$f = (f_n \circ \cdots \circ f_{n-r+1}) \circ (f_{n-r} \circ \cdots \circ f_1) := F \circ F_{n-r} \tag{9}$$

where $F_{n-r} = f_{n-r} \circ \cdots \circ f_1$ are layers to be *fixed* and $F = f_n \circ \dots \circ f_{n-r+1}$ are the last $r$ layers to be *finetuned* $(1 < r < n)$. With only two essential definitions above, our main theorem addressing the validity of the transfer learning technique can be derived:

**Theorem 3.5** (Finetuned loss bounded by pretrain loss and data). *Let $\mathcal{D}, \widetilde{\mathcal{D}}$ be two datasets with deviation $\varepsilon_{data}$ and $f = F \circ F_{n-r}$ be a well pretrained network under $\mathcal{D}$, then a net finetuning the last $r$ layers of $f$ with the form $g := (F + \delta F) \circ F_{n-r}$ has bounded loss on $\widetilde{\mathcal{D}}$,*

$$\mathcal{L}(g) = \frac{1}{N} \sum_{i=1}^{N} \|g(\widetilde{x}_i) - \widetilde{y}_i\|_{\mathcal{Y}}^2 \leq C_1 \varepsilon_{pretrained}^2 + C_2 \varepsilon_{data}^2, \tag{10}$$

*where $\delta F$ is a Lipschitz function with $C_{\delta F} \leq \varepsilon_{data}$. $C_1, C_2$ are two constants depending on $C_{F_{n-r}}$, $C_F$ and $C_{\widetilde{x}} := \max_i \|\widetilde{x}_i\|_{\mathcal{X}}$.*

*Sketch Proof.* By $g = (F + \delta F) \circ F_{n-r}$ and notations in Eq. (8), we have

$$g(\widetilde{x}_i) = (F + \delta F) \circ F_{n-r}(\widetilde{x}_i) = f(x_{j_i}) + v_1 \tag{11}$$

with $v_1 := \delta F \circ F_{n-r}(\widetilde{x}_i) + f(\widetilde{x}_i) - f(x_{j_i})$. We compute,

$$\frac{1}{N} \sum_{i=1}^{N} \|g(\widetilde{x}_i) - \widetilde{y}_i\|_{\mathcal{Y}}^2 = \frac{1}{N} \sum_{i=1}^{N} \|f(x_{j_i}) - \widetilde{y}_i + v_1\|_{\mathcal{Y}}^2 \leq 3 \left( \varepsilon_{pretrained}^2 + \varepsilon_{data}^2 + \|v_1\|_{\mathcal{Y}}^2 \right), \tag{12}$$

with Eq. (5) and the triangle inequality applied in the last inequality. Further, we estimate,

$$\|v_1\|_{\mathcal{Y}}^2 \leq 2 \Big( \|\delta F \circ F_{n-r}(\widetilde{x}_i))\|_{\mathcal{Y}}^2 + \|f(\widetilde{x}_i) - f(x_i)\|_{\mathcal{Y}}^2 \Big) \leq 2 \Big( C_{\delta F}^2 C_{F_{n-r}}^2 C_{\widetilde{x}}^2 + C_F^2 C_{F_{n-r}}^2 \varepsilon_{data}^2 \Big). \tag{13}$$

With Eq. (12) and condition $C_{\delta F} \leq \varepsilon_{data}$, the result Eq. (10) can be yielded. For the complete proof, see Appendix 7.1. $\qquad\square$

Theorem 3.5 depicts that the finetuned loss is bounded by the pretrained loss and the data difference in Eq. (10), linking the relation of *network functions* and *data* into one. This then explains our intuition why a well-pretrained net beforehand is necessary, and transfer learning is anticipated to be viable on datasets with certain similarities. Thus, with minimal conditions and definitions, Theorem 3.5 succinctly addresses when transfer learning guarantees a well-finetuned model under a new dataset. In fact, the generalization error under finetuning is also bounded (Theorem 7.1 Appendix).

**Remark 3.6.** *For two very different datasets, one has $\varepsilon_{data} \gg 0$. Eq. (10) then indicates such transfer learning process is generally more difficult as the upper bound becomes large, even with a small pretraining error $\varepsilon_{pretrained}$. Typically in domain adaptation tasks, $\varepsilon_{data}$ is not expected to be large.*

**Remark 3.7.** *Eq. (10) is regarded as an extension of Eq. (5) whenever $\widetilde{\mathcal{D}} \neq \mathcal{D}$. The two equations coincide if $\varepsilon_{data} = 0$ where two domains $\mathcal{D}, \widetilde{\mathcal{D}}$ perfectly align.*

## 4 INTERPRETATIONS & CHARACTERIZATIONS OF NETWORK-BASED TRANSFER LEARNING

Inspired by the mathematical framework of Theorem 3.5, an intuitive interpretation of network-based transfer learning can be derived, which in turn leads to a novel formulation for neural network finetuning. The derivation begins with the Layer Variational Analysis (LVA), where the effect of data variation and transmission can be addressed.

### 4.1 LAYER VARIATIONAL ANALYSIS (LVA)

To clarify the effect of one layer variation ($r = 1$ in Theorem 3.5), we denote the latent variables $z_i$, $\widetilde{z}_i$ and the last finetuned layer $g_n$ from Eq. (2) by,

$$z_i := F_{n-1}(x_i), \ \widetilde{z}_i := F_{n-1}(\widetilde{x}_i), \ g_n := f_n + \delta f_n \tag{14}$$

The target prediction error can be related to the pretrain error via the following expansion,

$$\|g(\widetilde{x}_i) - \widetilde{y}_i\|_{\mathcal{Y}} = \big\| \big(g_n(\widetilde{z}_i) - f_n(\widetilde{z}_i)\big) + \big(f_n(\widetilde{z}_i) - f_n(z_i)\big) + \big(f(x_i) - y_i\big) + \big(y_i - \widetilde{y}_i\big) \big\|_{\mathcal{Y}} \tag{15}$$

with six self-canceled-out terms added for auxiliary purposes. Note the identities $g(\widetilde{x}_i) = g_n(\widetilde{z}_i)$, $f_n(z_i) = f(x_i)$ are used and the second grouping can be rewritten by,

$$f_n(\widetilde{z}_i) - f_n(z_i) = J(f_n)(z_i) \cdot \delta z_i + \delta z_i^T \cdot H(f_n)(z_i) \cdot \delta z_i + \mathcal{O}(\delta z_i^3) \tag{16}$$

where $\delta z_i := (\widetilde{z}_i - z_i)$ corresponds to the *latent feature shift*; $J(f_n)(z_i)$ and $H(f_n)(z_i)$ are the *Jacobian* and the *Hessian* matrix of $f_n$ at $z_i$, respectively. In principle, non-linear activation functions introduce high order $\delta z_i$ terms to Eq. (16), where by considering regression type problems (assuming $f_n$ purely an affine function) the finetune loss Eq. (15) is allowed to be simplified as,

$$\mathcal{L}(g) = \frac{1}{N} \sum_{i=1}^{N} \|\delta f_n(\widetilde{z}_i) - q_i\|_{\mathcal{Y}}^2 \tag{17}$$

with $\delta y_i := \widetilde{y}_i - y_i$ and the vector $q_i \in \mathcal{Y}$ defined as

$$q_i := \delta y_i - J(f_n)(z_i) \cdot \delta z_i + (y_i - f(x_i)) \tag{18}$$

which is referred to as *transferal residue*. Via Transmission of Layer Variations (Appendix 7.3) one may also write,

$$q_i = \delta y_i - J(f)(x_i) \cdot \delta x_i + (y_i - f(x_i)) \tag{19}$$

## 4.2 LVA Interpretations & Intuitions

The equivalent form Eq. (17) of the transfer loss $\mathcal{L}(g) = \frac{1}{N} \sum_{i=1}^{N} \|g(\widetilde{x}_i) - \widetilde{y}_i\|_{\mathcal{Y}}^2$ under LVA renders an intuitive interpretation for network-based transfer learning. The resulting picture is particularly helpful for domain adaptations.

**Transferal residue:** We justify the name *transferal residue* of Eq. (19). There are two pairs:

$$q_i = \underbrace{\delta y_i - J(f)(x_i) \cdot \delta x_i}_{\text{domain mismatch } (\Delta)} + \underbrace{(y_i - f(x_i))}_{\text{pretrain error}(\star)} \tag{20}$$

where the first pair $(\Delta)$ evaluates *domain mismatch*; the other measures the *pretrain error*. To see this, observe: (a) a perfect pretrain $f$ gives $f(x_i) = y_i$ then $(\star) = 0$, and (b) if two domains match perfectly $\mathcal{D} = \widetilde{\mathcal{D}}$, $\delta x_i = \delta y_i = 0$ then $(\Delta) = 0$.

In such a perfect case, $q_i \equiv 0$ and there is *nothing new* for $f$ to learn/adapt in $\widetilde{\mathcal{D}}$. On the contrary, if $f$ is ill-pretrained with $\|\delta x_i\|, \|\delta y_i\|$ large, $q_i$ becomes large too, indicating there is *much to adapt* in $\widetilde{\mathcal{D}}$. Therefore, the transferal residue characterizes the amount of new knowledge needed to be learned in the new domain, and hence the name. In short,

$$\|q_i\| \to 0 \text{ (nothing to adapt)}, \quad \|q_i\| \to \infty \text{ (much to adapt)}$$

**Layer variations to cancel transferal residue:** Whenever the transferal residue is nonzero, the finetune net $g$ adjusts from $f$ to react, as Eq. (17) assigns layer variation $\delta f_n$ to neutralize $q_i$. Minimizing the adaptation loss $\mathcal{L}(g)$ with $\delta f_n$, we see that

$$q_i \approx 0 \xrightarrow[\text{adapt}]{no} \delta f_n \approx 0, \qquad q_i \neq 0 \xrightarrow[\text{adapt}]{need} \delta f_n \overset{\otimes}{\Longleftarrow} \min \text{Eq. (17)}$$

In fact, the last adaptation step $\otimes$ is analytically solvable if $\delta f_n$ is a linear functional with $\|\cdot\|_{\mathcal{Y}}$ being $L_2$-norm. By Moore-Penrose pseudo-inverse, the minimum is achieved at

$$\delta f_n = \left(\widetilde{z}^T \cdot \widetilde{z}\right)^{-1} \widetilde{z}^T \cdot q \tag{21}$$

with $\tilde{z} := (\widetilde{z}_1, \ldots, \widetilde{z}_N)$ and $q := (q_1, \ldots, q_N)$. In short, layer variations tend to digest the knowledge discrepancy due to domain differences.

**Knowledge transfer:** As $q_i$ reflects the amount of new knowledge left to be digested in $\widetilde{\mathcal{D}}$, we see a term $J(f)(x_i)\delta x_i$ automatically emerges trying to reduce the remainder in $q_i$. Especially, the Jacobian term appears with a negative sign opposite to $f(x_i)$; it is presented as a *self-correction (self-adaptation)* to the new domain. The purpose of $J(f)(x_i)\delta x_i$ serves to **annihilate the new label shift** $\delta y_i$ Eq. (20) (e.g. Fig. 1) and notice that the self-correction not needed if $\mathcal{D} = \widetilde{\mathcal{D}}$, $\delta x_i = 0$.

Since the pretrained $f$ contains previous domain information, the term $J(f)(x_i)\delta x_i$ indicates the **old knowledge is carried over by** $f$ to dissolve the new unknown data. Such quantification then

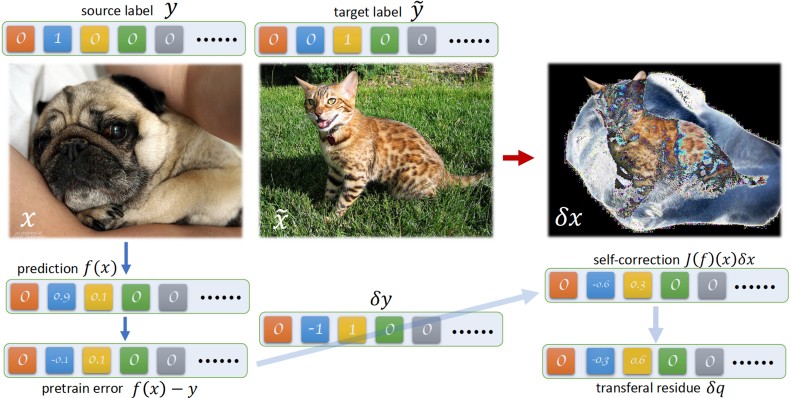

Figure 1: An illustration of transferal residue Eq. (20). In this example, a pretrained (animal classifier) $f$ with 1-hot label is adapted from $\mathcal{D} \to \widetilde{\mathcal{D}}$, where $\mathcal{D}$ is of dog images and $\widetilde{\mathcal{D}}$ is of cat images. The self-correction term transferring the old knowledge attempts to correct predictions, Eq. (20).

yields an interpretation and renders an intuitive view for network-based transfer learning. The LVA formulation naturally admits 4 types of domain transitions according to Eq. (20): (a) $\delta x_i = \delta y_i = 0$, (b) $\delta x_i \neq 0, \delta y_i = 0$, (c) $\delta x_i = 0, \delta y_i \neq 0$ and (d) $\delta x_i \neq 0, \delta y_i \neq 0$.

The analysis can be extended to multi-layer cases as well as Convolutional Neural Networks (CNN); only that the analytic computation soon becomes cumbersome. Therefore, only one-layer LVA finetuning is demonstrated in this work. Further discussions and calculations for LVA of two layers and CNNs can be found in Appendix 7.2. Although there is no explicit formula for multi-layer cases to obtain the optimal solution, iterative regression can be utilized recursively to obtain a sub-optimal result analytically without invoking gradient descent. The LVA formulation not only renders interesting perspectives in theory but also demonstrates successful knowledge adaptations in practice.

## 5 EXPERIMENTS

Three experiments in various fields were conducted to verify our theoretical formulation. The first task was a simulated 1D time series regression, the second was Speech Enhancement (SE) of a real-world voice dataset; the third was an image deblurring task by extending our formula onto Convolutional Neural Networks (CNNs). The three tasks demonstrated the LVA had prompt adaptation ability under new domains and confirmed the calculations Eq. (15)~(18). The code is available on Github[1]. For detailed implementations and additional results on adaptive image classifications, see Appendix 8.

### 5.1 TIME SERIES (1D SIGNAL) REGRESSION

**Goal:** Train a DNN predicting temporal signal $\mathcal{D}$ and adapt it to another signal with noise $\widetilde{\mathcal{D}}$.

**Dataset:** Two signals are designed as follows, with $\widetilde{\mathcal{D}}$ mimicking a noisy signal (a low-quality device)

$$\mathcal{D} = \big\{ (t_i, \sin(5\pi t_i)) \big\}_i, \quad \widetilde{\mathcal{D}} = \big\{ (t_i + 0.05\,\xi_i, \gamma(t_i)\sin(5\pi t_i)) + 0.03\,\eta_i) \big\}_i \qquad (22)$$

with $\{t_i \,|\, i = 1, \ldots, 2000\} \in [-1, 1]$ equally distributed, $\gamma(t_i) = 0.4\,t_i + 1.3998$, $\xi_i \sim \mathcal{N}(1.5, 0.8)$, $\eta_i \sim \mathcal{U}(-1, 1)$. $\mathcal{N}(\mu, \sigma)$ is a normal distribution of mean $\mu$ and variance $\sigma$; $\mathcal{U}$ as a uniform distribution $[-1, 1]$.

**Implementation:** The pretrained model $f$ consists of 3 fully-connected layers of 64 nodes with ReLu as activation functions. We refer to Appendix for more implementation details.

**Result analysis:** The adaptation results were compared between the conventional gradient descent (GD) method and LVA by Eq. (18), (21). The transferred net $g$ from $f$ was finetuned with GD for 12000 epochs, while LVA only requires one step to derive. After adaptation, LVA reached the lowest $L_2$-loss compared to GD of same finetuned layer numbers, see Fig. 2(f). This verified the stability and validity of the proposed framework. In addition to obtaining the lowest loss, the actual signal

---

[1] https://github.com/HHTseng/Layer-Variational-Analysis.git

prediction performance can also be observed improved via visualization Fig. 2(b)∼(e) in the case of finetuning the last one and two layers. Notably, the finetuning loss of two layers indeed further decreased than that of one layer, which met the expectation as the two-layer case had more free parameters to reduce loss. Comparing Fig. 2(b)(c), it showed that the 1-layer case LVA obtained more accurate signal predictions compared to GD's predictions. Fig. 2(d)(e) showed the results of 2-layer finetune by GD and LVA. The 2-layer LVA formulation is derived in Appendix 7.2. Observing that the LVA formulation equips prompt adaptation ability, we further conduct two real-world applications.

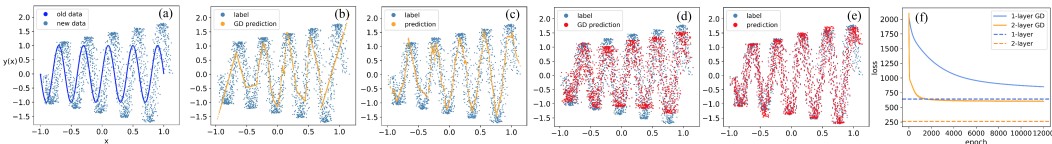

Figure 2: (a) Datasets $\mathcal{D}$ and $\widetilde{\mathcal{D}}$ given by Eq. (22), (b) 1-layer finetuning by GD, (c) 1-layer LVA finetuning, (d) 2-layer finetuning by GD, (e) 2-layer LVA finetuning, (f) the training loss on the finetuned net over epochs.

## 5.2 SPEECH ENHANCEMENT

To prove that the proposed method is effective on real data, we conducted experiments on an SE task.

**Goal:** Train a SE model to enhance (denoise) speech signals in a noisy environment $\mathcal{D}$ and adapt to another noisy environment $\widetilde{\mathcal{D}}$.

**Dataset:** Speech data pairs were prepared for the source domain $\mathcal{D}$ (served as the training set) and target domain $\widetilde{\mathcal{D}}$ (served as the adaptation set) as follows:

$$\mathcal{D} = \{(x_{ijk}, y_i)\}_{i,k}^{N_1, N_2} \xrightarrow{\text{adapt}} \widetilde{\mathcal{D}} = \{(\widetilde{x}_{ij}, \widetilde{y}_i)\}_{i,k}^{\widetilde{N}_1, \widetilde{N}_2} (x_{ijk} = y_i + c_j \times n_k; \widetilde{x}_{ijk} = \widetilde{y}_i + \widetilde{c}_j \times \widetilde{n}_k)$$

where $y_i$ denotes a patch[2] of a clean speech (as a label), which is then corrupted with noise $n_k$ of type $k$ to form a noisy speech (patch) $x_{ijk}$ over an amplification of $c_j \propto \exp(-\text{SNR}_j)$, determined by a signal-to-noise ratio ($\text{SNR}_j$) given. $N_1$ (*resp.* $\widetilde{N}_1$) denotes the number of patches extracted from clean utterances in $\mathcal{D}$ (*resp.* $\widetilde{\mathcal{D}}$); $N_2$ (*resp.* $\widetilde{N}_2$) is the number of noise types contained in $\mathcal{D}$ (*resp.* $\widetilde{\mathcal{D}}$). An SE model $f$ on $\mathcal{D}$ performed denoising such that $f(x_{ijk}) = y_i$. In this experiment, 8,000 utterances (corresponding to $N_1 = 112{,}000$ patches) were randomly excerpted from the Deep Noise Suppression Challenge (Reddy et al., 2020) dataset; these clean utterances were further contaminated with domain noise types: $n_k \in \{$White-noise, Train, Sea, Aircraft-cabin, Airplane-takeoff$\}$, (thus $N_2 = 5$), at four SNR levels: $\{-5, 0, 5, 10\}$ (dB) to form the training set. These 112,000 noisy-clean pairs were used to train the pretrained SE model. We tested performance using different numbers of adaptation data with $\widetilde{N}_1$ varying from 20 to 400 patches (Fig. 3). These adaptation data were contaminated by three target noise types: $\widetilde{n}_k \in \{$Baby-cry, Bell, Siren$\}$ (thus $\widetilde{N}_2 = 3$) at $\{-1, 1\}$ (dB) SNR. Note the speech contents, noise types, and SNRs were all mismatched in the training and adaptation sets.

**Implementation:** A Bi-directional LSTM (BLSTM) (Chen et al., 2015) model was used to construct the SE model $f$ under $L_2$-loss, consisting of one-layer BLSTM of 300 nodes and one fully-connected layer for output. The transferred net $g$ adapted from $f$ by finetuning the last layer was performed by GD to compare with LVA in Eq. (21). For data processing and network, details see Appendix 8.2.

**Domain alignment:** It was mentioned that the domain alignment ought to be applied prior to LVA. For real-world data, *optimal transport* (Villani, 2009) (OT) was selected for implementation to match domain distributions. To contrast, LVA with and without OT were conducted.

**Evaluation metric:** SE performances are usually measured by 3 evaluation metrics, $L_2$-error (MSE), the perceptual evaluation of speech quality (PESQ) (Rix et al., 2001) and short-time objective intelligibility (STOI) (Taal et al., 2011). Each metric evaluates different aspects of speech: PESQ $\in [-0.5, 4.5]$ evaluates speech quality; STOI $\in [0, 1]$ estimates speech intelligibility.

---

[2]A patch is defined as a temporal segment extracted from an utterance. We used 128 *magnitude spectral* vectors to form a patch in practice.

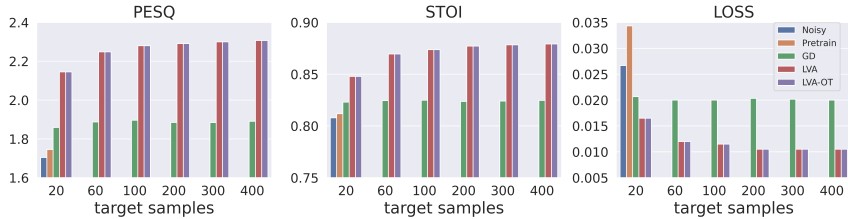

Figure 3: Performance of GD vs. LVA by different patch amount $\widetilde{N}_1$ of $\widetilde{\mathcal{D}}$ for adaptation.

**Result analysis:** We excerpted another 100 utterances, contaminated with the target noise types: $\widetilde{n}_k \in \{$Baby-cry, Bell, Siren$\}$ at $\{-1, 1\}$ (dB) SNR levels, to form the test set. Note that the speech contents were mismatched in the adaptation and test sets. Fig. 3 showed the domain adaptation results from environment $\mathcal{D} \to \widetilde{\mathcal{D}}$ to compare GD with LVA. On this test set, the PESQ and STOI of the original noisy speech (without SE) were 1.704 and 0.808, respectively. The results in Fig. 3 showed consistent out-performance of LVA over GD under a different number of adaptation data in $\widetilde{\mathcal{D}}$ ($\widetilde{N}_1$ as the horizontal axis). Especially, the $L_2$-loss of LVA was notably less than that of GD, confirming that LVA indeed derived the globally optimal weights of a transferred net. Meanwhile, it was observed that LVA significantly outperformed GD when the amount of target samples in $\widetilde{\mathcal{D}}$ was considerably less. It was also noticed that the LVA equipped OT alignment (LVA-OT) achieves similar performance to LVA. An example of the enhanced spectrograms by different SE models: the Pretrained, GD, LVA, and LVA-OT are shown in Fig. 4 with clean and unprocessed noisy spectrograms for comparison. Fig. 4 shows LVA recovers the speech components with clear structures than Pretrained and GD (see green rectangles). More extensive result analyses are provided in Appendix 8.2.

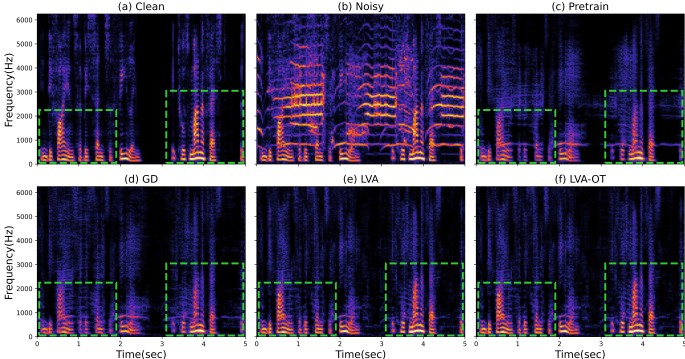

Figure 4: Spectrograms of an utterance (Baby-cry noise at SNR $-1$ dB; $\widetilde{N}_1 = 400$ for adaptation).

## 5.3 SUPER RESOLUTION FOR IMAGE DEBLURRING

**Goal:** Train a Super-Resolution Convolutional Neural Network (SRCNN, Fig. 5) (Dong et al., 2016) to deblur images of domain $\mathcal{D}$ and adapt to another *more blurred* domain $\widetilde{\mathcal{D}}$.

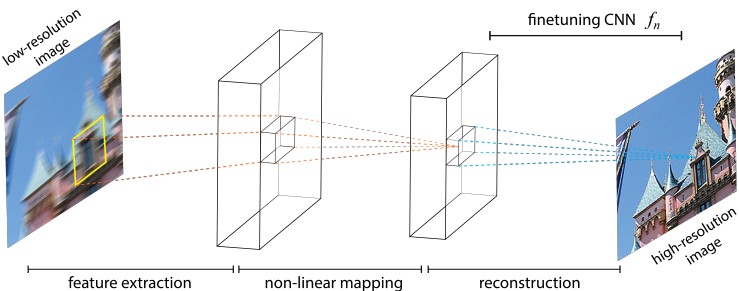

Figure 5: [SRCNN for image deblurring] Pretrain model: a sequence of CNN layers $f = f_n \circ \cdots f_2 \circ f_1$. Then last CNN layer $f_n$ is finetuned to adapt to more blurred images in $\widetilde{\mathcal{D}}$.

**Dataset:** 2000 high-resolution images were randomly selected as labels from the CUFED dataset (Wang et al., 2016) to train SRCNN. The source domain $(x, y) \in \mathcal{D}$ and the target domain $(\widetilde{x}, \widetilde{y}) \in \widetilde{\mathcal{D}}$ were set as follows,

$$x \in \{\times 3 \text{ blurred patches}\}, \ \widetilde{x} \in \{\times 4, \times 6 \text{ blurred patches}\}, \ y = \widetilde{y} \in \{\text{High-Res patches}\}$$

Fig. 6 shows image pairs for adaptation. Another image dataset SET14 (Zeyde et al., 2010) was used to test adapted models. More image preparation details see Appendix 8.3.

| $y$: original | $x$: bicubic x3 | $\widetilde{x}$: bicubic x4 | $\widetilde{x}$: bicubic x6 |
| --- | --- | --- | --- |

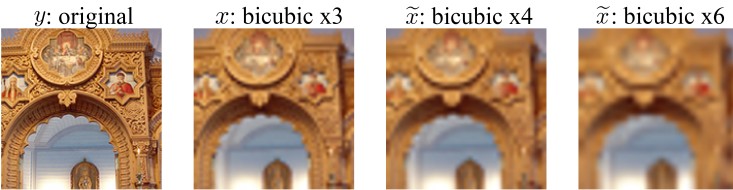

Figure 6: Sample images (CUFED) of $\mathcal{D}$ and $\widetilde{\mathcal{D}}$.

**Implementation:** This experiment demonstrated that image related tasks can also be implemented by extending our LVA, Eq. (17), (18), from fully-connected layers to CNNs. Although the analytic formula Eq. (21) is only valid on a fully-connected layer of regression type, a key observation admitting such extension is that CNN kernels can locally be regarded as a fully-connected layer, in that every receptive field is exactly covered by the kernels of a CNN. By proper arrangement, a CNN kernel $\mathcal{C} = (\mathcal{C}_{ij\gamma})$ can be folded as a fully-connected layer of 2D matrix weight $W = (W_{\ell m})$. As such, the LVA can be extended onto CNNs as finetune layers; detailed implementations see Appendix. In this experiment, an SRCNN model $f$ (Fig. 5) was trained on source domain $\mathcal{D}$ to perform deblur $f(x) = y$, and subsequently used GD and LVA to adapt $f \to g$ on $\widetilde{\mathcal{D}}$ such that $g(\widetilde{x}) = \widetilde{y}$.

**Result analysis:** Fig. 7 showed the results of adapted models by GD and LVA. In Fig. 7, column (b) were more blurred inputs $\widetilde{x}$ of $\widetilde{\mathcal{D}}$; column (c)(d)(e) were deblurred images to be compared with the ground truths in column (a). Column (c)(d) compared the two methods adapted with the same number of target domain samples ($N = 256$ patches) and LVA reached the highest PSNR scores. While more target domain samples were exclusively included for GD to $N = 16384$ in column (e), the corresponding PSNRs were still not compatible to the LVA adaptation. There were more extensive experiments conducted to contrast the adaptation outcome of GD and LVA, see Appendix 8.3.

| (a) ground truth | (b) bicubic x6 | (c) LVA (N=256) | (d) GD (N=256) | (e) GD (N=16384) |
| --- | --- | --- | --- | --- |

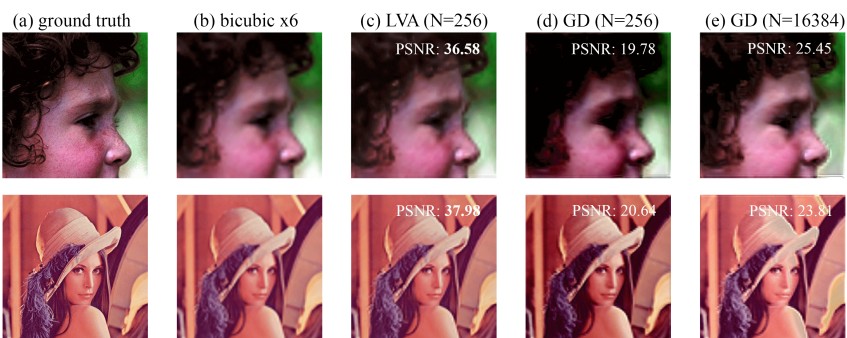

Figure 7: Image deblurring results of adapted SRCNNs on testing data SET14.

## 6 CONCLUSIONS

This study aims to provide a theoretical framework for interpreting the mechanism under domain adaptation. We derived a firm basis to ensure that a finetuned net can be adapted to a new dataset. The objective of this study was achieved to address "why transfer learning might work?" and unravel the underlying process. Based on the LVA, a novel formulation for finetuned nets was introduced and yielded meaningful interpretations of transfer learning. The formalism was further validated by domain adaptation applications in Speech Enhancement and Super Resolution. Various tasks were observed to reach the lowest $L_2$-loss under LVA, where the LVA is the theoretical limit for the gradient descent to approach. The LVA interpretations and successful experiments give this study an inspiring perspective to view transfer learning.

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

# 7 APPENDIX

## 7.1 PROOF OF THEOREM 3.6

Consider finetuning on the last $r$ layers in Eq. (2) with $k + r = n$. Denote $F_{n-r} = f_{n-r} \circ \cdots \circ f_1$ and the last $r$ layers $F = f_n \circ ... \circ f_{n-r+1}$ of pretrained network and finetuned net $G = g_n \circ ... \circ g_{n-r+1}$. We also write the difference $G - F = \delta F$. With the notations, we rewrite

$$g(\widetilde{x}_i) = G \circ F_{n-r}(\widetilde{x}_i) = (F + \delta F) \circ F_{n-r}(\widetilde{x}_i) = f(x_{j_i}) + v_1, \tag{A.23}$$

where $v_1 = \delta F \circ F_{n-r}(\widetilde{x}_i) + f(\widetilde{x}_i) - f(x_{j_i})$ and $j_i$ is the corresponding index such that Eq. (7) is satisfied. Therefore, one computes,

$$
\begin{aligned}
\frac{1}{N} \sum_{i=1}^{N} \|g(\widetilde{x}_i) - \widetilde{y}_i\|_{\mathcal{Y}}^2 &= \frac{1}{N} \sum_{i=1}^{N} \|f(x_{j_i}) - \widetilde{y}_i + v_1\|_{\mathcal{Y}}^2 \\
&\leq \frac{3}{N} \sum_{i=1}^{N} \left( \|f(x_{j_i}) - y_{j_i}\|_{\mathcal{Y}}^2 + \|y_{j_i} - \widetilde{y}_i\|_{\mathcal{Y}}^2 + \|v_1\|_{\mathcal{Y}}^2 \right) \\
&\leq 3 \left( \varepsilon_{\text{pretrained}}^2 + \varepsilon_{\text{data}}^2 + \|v_1\|_{\mathcal{Y}}^2 \right),
\end{aligned}
\tag{A.24}
$$

where the last term is further estimated by the triangle inequality to yield

$$
\begin{aligned}
\|v_1\|_{\mathcal{Y}}^2 &\leq 2 \Big( \|\delta F \circ F_{n-r}(\widetilde{x}_i))\|_{\mathcal{Y}}^2 + \|f(\widetilde{x}_i) - f(x_{j_i})\|_{\mathcal{Y}}^2 \Big) \\
&\leq 2 \Big( C_{\delta F}^2 \, C_{F_{n-r}}^2 \, C_{\widetilde{x}}^2 + C_F^2 \, C_{F_{n-r}}^2 \, \varepsilon_{\text{data}}^2 \Big).
\end{aligned}
\tag{A.25}
$$

with $C_{\widetilde{x}} := \max_i \|\widetilde{x}_i\|_{\mathcal{X}}$. Note that without any additional assumption, the first term in Eq. (A.25) is of order 1. To obtain a small error of $g$, one sufficient condition is to have $C_{\delta F} \approx \varepsilon_{\text{data}}$. It is interesting to point out that the analysis relates the Lipschitz constant $C_{\delta F}$ of the network perturbation with the data similarity $\varepsilon_{\text{data}}$.

**Theorem 7.1** (Generalization error bound). *Given a test (held-out) set $\widetilde{D}_{test} = \{(x_i^{(test)}, y_i^{(test)}) \in \mathcal{X} \times \mathcal{Y}\}_{i=1}^{N_{test}}$ with $|\widetilde{D}_{test}| \le |\widetilde{D}|$, then a well-finetuned net $g$ in Theorem 3.5 has a generalization error bound,*

$$\mathcal{L}_{\text{test}}(g) = \frac{1}{N_{\text{test}}} \sum_{i=1}^{N_{\text{test}}} \|g(\widetilde{x}_i^{\text{test}}) - \widetilde{y}_i^{\text{test}}\|_{\mathcal{Y}}^2 \le C_1\, \varepsilon_{\text{data}}^2 + C_2\, \mathcal{L}(g) \tag{A.26}$$

where $\varepsilon_{\text{data}}$ is the data deviation between $\widetilde{D}$ and $\widetilde{D}_{\text{test}}$ computed by Definition 3.4. $C_1$, $C_2$ are two constants with $C_1$ depending on the Lipschitz constant $C_g$ and $\mathcal{L}(g)$ the finetune loss in Theorem 3.5.

*Proof.*

$$\begin{aligned}
\mathcal{L}_{\text{test}}(g) &= \frac{1}{N_{\text{test}}} \sum_{i=1}^{N_{\text{test}}} \|g(\widetilde{x}_i^{\text{test}}) - \widetilde{y}_i^{\text{test}}\|_{\mathcal{Y}}^2 \\
&\le \frac{3}{N_{\text{test}}} \sum_{i=1}^{N_{\text{test}}} \left( \|g(\widetilde{x}_i^{\text{test}}) - g(\widetilde{x}_i)\|_{\mathcal{Y}}^2 + \|\widetilde{y}_i^{\text{test}} - \widetilde{y}_i\|_{\mathcal{Y}}^2 + \|g(\widetilde{x}_i) - \widetilde{y}_i\|_{\mathcal{Y}}^2 \right) \\
&\le \frac{3}{N_{\text{test}}} \sum_{i=1}^{N_{\text{test}}} \left( C_g^2\, \varepsilon_{\text{test}}^2 + \varepsilon_{\text{test}}^2 \right) + \frac{3}{N_{\text{test}}} \sum_{i=1}^{N} \|g(\widetilde{x}_i) - \widetilde{y}_i\|_{\mathcal{Y}}^2
\end{aligned} \tag{A.27}$$

$\square$

## 7.2 PERTURBATION APPROXIMATION DERIVATION FOR 2-LAYER CASE

Similarly, let $\delta z_i := \widetilde{z}_i - z_i$ and denote the finetuned $n^{\text{th}}$ layer and $(n-1)^{\text{th}}$ layer by

$$g_n = f_n + \delta f_n, \quad g_{n-1} = f_{n-1} + \delta f_{n-1} \tag{A.28}$$

The perturbation assumption allows the approximation of

$$f_{n-1}(\widetilde{z}_i) \approx f_{n-1}(z_i) + J(f_{n-1})(z_i) \cdot \delta z_i, \tag{A.29}$$

where $J(f_{n-1})(z_i)$ is the Jacobian matrix of $f_{n-1}$ at $z_i$. The network assumed to deviate from the pretrained $f_{n-1}$ by an affine function,

$$\delta f_{n-1}(\widetilde{z}_i) := W_{n-1} \cdot \delta z_i + b, \tag{A.30}$$

for some weight $W_{n-1}$ and bias $b$. With Eq. (A.29) and Eq. (A.30), we obtain

$$g_{n-1}(\widetilde{z}_i) \approx f_{n-1}(z_i) + b + \left( J(f_{n-1})(z_i) + W_{n-1} \right) \cdot \delta z_i. \tag{A.31}$$

With Eq. (A.31),

$$\begin{aligned}
g(\widetilde{x}_i) &= (f_n + \delta f_n) \circ g_{n-1}(\widetilde{z}_i) \\
&\approx f_n \circ f_{n-1}(z_i) + f_n \circ \left( \left( J(f_{n-1})(z_i) + W_{n-1} \right) \cdot \delta z_i + b \right) + \delta f_n \circ f_{n-1}(z_i).
\end{aligned} \tag{A.32}$$

Note that the terms involving multiplications $\delta f_n$ and $\delta f_{n-1}$ are omitted as we only focus on the first order terms. Parallel to Eq. (15) for 1-layer case, we approximate

$$\|g(\widetilde{x}_i) - \widetilde{y}_i\|_{\mathcal{Y}} = \| (g(\widetilde{x}_i) - f_n \circ f_{n-1}(z_i)) + (f_n \circ f_{n-1}(z_i) - y_i) + (y_i - \widetilde{y}_i) \|_{\mathcal{Y}} \tag{A.33}$$

and therefore,

$$\|g(\widetilde{x}_i) - \widetilde{y}_i\|_{\mathcal{Y}} \approx \|\delta f_n \circ f_{n-1}(z_i) + f_n \circ \left( \left( J(f_{n-1})(z_i) + W_{n-1} + b \right)(\widetilde{z}_i - z_i) \right) + f(x_i) - y_i - \delta y_i\|_{\mathcal{Y}}. \tag{A.34}$$

We see that the 2-layer fine-tuning case involves three unknown (parameters), including $\delta f_n$, $b$, and $W_{n-1}$. Although there is no explicit formula to obtain the optimal solution, iterative regression can be utilized multiple times to obtain a sub-optimal result analytically without using gradient descent. In fact, one notices that if $b$ is ignored, the optimization of $\delta f_n$ and $W_{n-1}$ based on the $L_2$-loss can be solved. Once $\delta f_n$ and $W_{n-1}$ are solved and fixed, $b$ can be introduced back into the formula and can be solved again by regression. Similar approximations for general multi-layers cases would still be reliable as long as the features in the latent space are close to each other. That is to say, the features of the source and target domain share similarities. These intuition-motivated formulations were shown to successfully adapt knowledge in experiments for both images and speech signals. The results can be found in Section 5.

### 7.3 Transmission of Layer Variations

We show heuristically the propagation of layer variations in neural nets can be traced via data deviations. Given two datasets $\mathcal{D}$, $\widetilde{\mathcal{D}}$ with the indexing after sample alignment, we define

$$\delta x_i := \widetilde{x}_i - x_i, \quad \delta y_i := \widetilde{y}_i - y_i \tag{A.35}$$

Let the data deviation be $\varepsilon$ from Def. 3.4, and Eq. (A.35) can be decomposed by orders of $\varepsilon$,

$$\delta x_i = \delta x_i^{(1)} + \delta x_i^{(2)} + \cdots, \quad \delta \widetilde{y}_i = \delta y_i^{(1)} + \delta y_i^{(2)} + \cdots \tag{A.36}$$

with $\mathcal{O}\left(\|\delta x_i^{(k)}\|_{\mathcal{X}}\right) = \mathcal{O}\left(\|\delta y_i^{(k)}\|_{\mathcal{Y}}\right) = \varepsilon^k$. To obtain a simple picture, first we neglect the high order terms $\delta x_i^{(2)}, \delta y_i^{(2)}$, etc. In such case, $\delta x_i^{(1)}$ and $\delta y_i^{(1)}$ can be simply denoted as $\delta x_i$ and $\delta y_i$, respectively, without confusion. By Eq. (14), finetuning the last layer in Theorem 3.5 reveals that

$$\begin{aligned}
\mathcal{L}(g) &= \frac{1}{N} \sum_{i=1}^{N} \| (f_n + \delta f_n) \circ f_{n-1} \circ \cdots \circ f_1 (\widetilde{x}_i) - \widetilde{y}_i \|_{\mathcal{Y}}^2 \\
&= \frac{1}{N} \sum_{i=1}^{N} \| \delta f_n(\widetilde{z}_i) - \widetilde{y}_i + f(x_i) + J(f)(x_i)\delta x_i \|_{\mathcal{Y}}^2 + \mathcal{O}(\varepsilon^2)
\end{aligned} \tag{A.37}$$

where $J(f)(x_i)$ is the Jacobian of $f$ at $x_i$ and $\mathcal{O}(\varepsilon^2)$ contains terms like $H(f)(x_i)\delta x_i \delta x_i$, etc with $H(f)(x_i)$ the *Hessian* tensor of $f$ at $x_i$. Some effort converts Eq. (A.37) into a familiar form,

$$\mathcal{L}(g) = \frac{1}{N} \sum_{i=1}^{N} \| \delta f_n(\widetilde{z}_i) - q_i \|_{\mathcal{Y}}^2 \tag{A.38}$$

with

$$q_i = \delta y_i - J(f)(x_i) \cdot \delta x_i + (y_i - f(x_i)) \tag{A.39}$$

Regard that Eq. (A.38) recovers Eq. (16) via different routes and starting points, although the two $q_i$'s in Eq. (18) and (A.39) slightly differ due to distinct variational conditions. This heuristic derivation then confirms that data deviations arouse network variations to induce the alternative finetune form. Consequently, both derivations yield same results.

## 8 Experimental Details

All code and data can be found at `https://github.com/HHTseng/Layer-Variational-Analysis.git`. The norms $\| \cdot \|_{\mathcal{X}}$ and $\| \cdot \|_{\mathcal{Y}}$ in all experiments are set as Euclidean $L_2$-norms, unless otherwise specified. Some notable details are remarked here.

## 8.1  1D SERIES REGRESSION

**Dataset**  A 1D series $\mathcal{D}$ for the pretrained $f$ and another series $\widetilde{\mathcal{D}}$ for transfer learning are given by (Fig. 8)

$$\mathcal{D} = \left\{ (x_i, \sin(5\pi x_i)) \,|\, x_i \in [-1, 1] \right\}_{i=1}^{N=2000}$$

$$\widetilde{\mathcal{D}} = \left\{ \underbrace{(x_i + 0.05\, \xi_i, \gamma(x_i)\, y_i + 0.03\, \eta_i)}_{(\widetilde{x}_i, \widetilde{y}_i)} \,|\, x_i \in [-1, 1] \right\}_{i=1}^{N=2000} \tag{B.40}$$

with

$$\gamma(x_i) = 0.4\, x_i + 1.3998,$$
$$\xi_i \sim \mathcal{N}(1.5, 0.8) \quad \text{(normal distribution mean=1.5, variance=0.8),}$$
$$\eta_i \sim \mathcal{U}(-1, 1) \quad \text{(uniform distribution in } [-1, 1]).$$

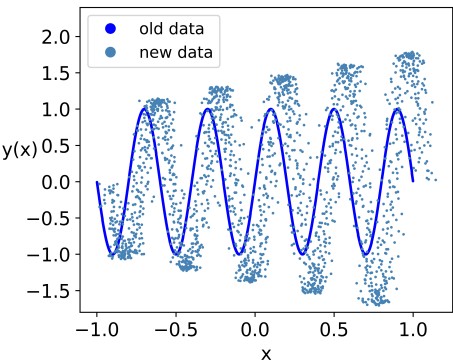

Figure 8: The original data $\mathcal{D}$ and new data $\widetilde{\mathcal{D}}$ for transfer learning.

**Network Architecture**  The pretrained model $f$ using $\mathcal{D}$ was a 3 fully-connected-layer network with 64 nodes at each layer and ReLu used as activation functions except the output layer. A finetuned model $g_{\text{GD}}$ using Gradient Descent (GD) retrained the last layer of $f$ by $\widetilde{\mathcal{D}}$. $f$ and $g_{\text{GD}}$ were trained under $L_2$-loss with ADAM optimizer at learning rate $10^{-3}$ by 8,000 and 12,000 epochs, respectively. The finetuned model by the LVA method $g_{\text{pesudo}}$ using $\widetilde{\mathcal{D}}$ required no training process but directly replaced the last layer of $f$ by $f_n \mapsto (f_n + \delta f_n)$ with $\delta f_n$ given in Eq. (21).

**Hardware**  One NVIDIA V100 GPU (32 GB GPU memory) with 4 CPUs (128 GB CPU memory).

**Runtime**  Approximately 3 sec in the LVA method, 4 minutes in GD method.

## 8.2  SPEECH ENHANCEMENT

**Dataset**  The utterances used in the experiment were excerpted from the Deep Noise Suppression Challenge (Reddy et al., 2020). We randomly selected 8000 clean utterances for training and 100 utterances for testing. Five noise types {White-noise, Train, Sea, Aircraft-cabin, Airplane-takeoff} were involved to form the training set (to estimate the pretrained SE model) and three noise types, {Baby-cry, Bell, Siren} was used to form the transfer learning and testing sets. For the training set, the 8000 clean utterances were equally divided and contaminated by the five noise types (thus, each noise type had 1600 noisy utterances) at four signal-to-noise ratio (SNR) levels, $\{-5, 0, 5, 10\}$ dB. For the testing set, 100 clean utterances were contaminated by the {Baby-cry, Bell, Siren} noises at $\{-1, 1\}$ (dB) SNRs. For the adaptation set, we prepared 20-400 patches contaminated by three noise types, {Baby-cry, Bell, Siren}, at $\{-1, 1\}$ dB SNRs.

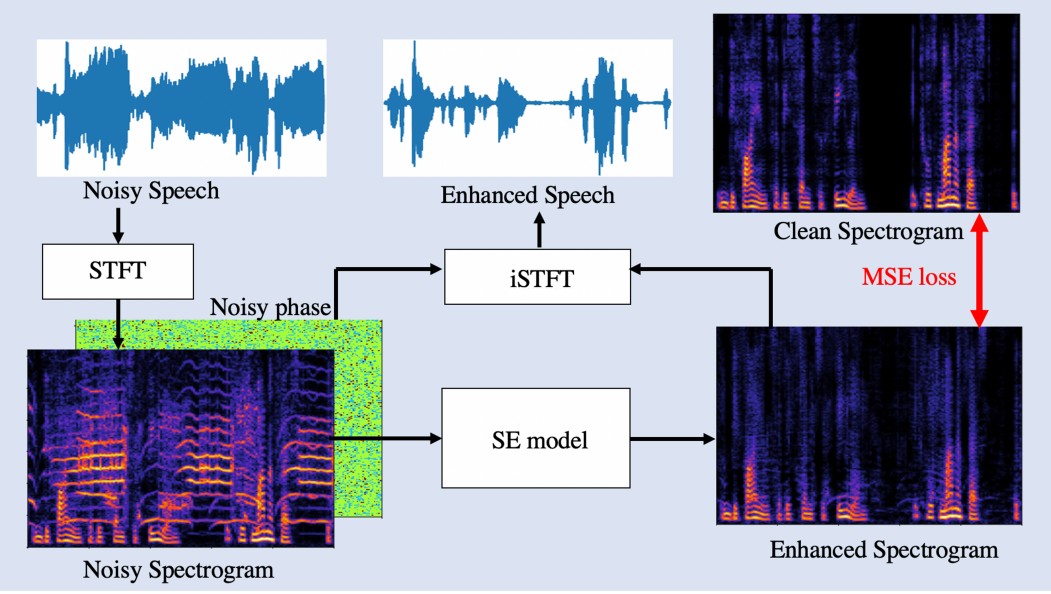

Figure 9: The flowchart of a Deep Denoising Autoencoder (DDAE) for SE.

**Implementation**   All speech utterances were recorded in a 16kHz sampling rate. The speech signals were first transformed to spectral features (magnitude spectrograms) by a short-time Fourier transform (STFT). The Hamming window with 512 points and a hop length of 256 points were applied to transform the speech waveforms into spectral features. To adequately shrink the magnitude range, a log1p function ($log1p(x) := log(1 + x)$) was applied to every element in the spectral features. During training, noisy spectral vectors were treated as input to the SE model to generate enhanced spectral vectors. The clean spectral vectors were treated as the reference, and the loss was computed based on the difference of enhanced and reference spectral vectors. Subsequently, the SE model was trained to minimize the difference. During inference, noisy spectral vectors were inputted to the trained SE model to generate enhanced ones, which were then converted into time-domain waveform with the preserved original noisy phase via inverse STFT.

**Network Architecture**   The overall flowchart of the SE task is shown in Fig. 9. In this study, we implemented SE systems using two neural network models to evaluate the proposed LVA. The first one is the deep denoising autoencoder (DDAE) model  (Lu et al., 2013), which serves as a simpler neural-network architecture. The other is the Bi-directional LSTM (BLSTM)  (Chen et al., 2015), which is relatively complicated as compared to DDAE. For DDAE, the model is composed of 5 fully-connected layers with 512 units. Activation functions were added to each layer except the last layer. The LeakyReLU with a negative slope of 0.01 was used as the activation function. For BLSTM, the model consists of a one-layer BLSTM of 300 nodes and one fully-connected layer. The training setting is the same for the two SE models. The learning rate of Adam Optimizer was set at $10^{-3}$. After pretraining, only the last layer parameters were updated in the finetuning stage. According to Eq. (21), the pseudo inverse of latent features was used to calculate $\delta f$.

**Hardware**   One NVIDIA V100 GPU (32 GB GPU memory) with 4 CPUs (128 GB CPU memory).

**Runtime**   Approximately 7 minutes in the LVA method and 25 minutes in GD method with 40 utterances.

### 8.2.1   ADAPTATION ON MISMATCHED SPEAKERS

In Sec. 5.2, we have confirmed the effectiveness of the proposed LVA on the noise type and SNR adaptation of the SE task, where the main speech distortion is from the background noise. In

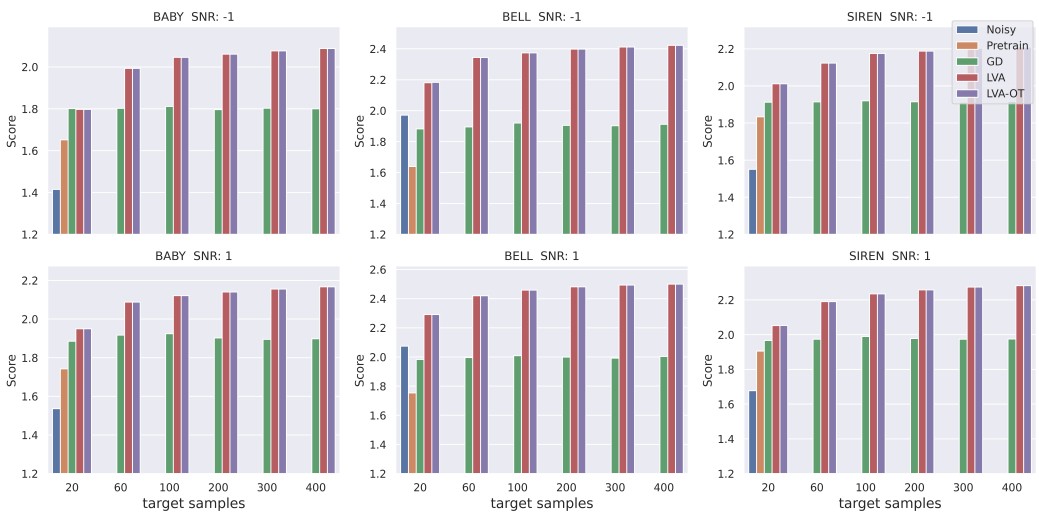

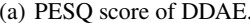

(a) PESQ score of DDAE.

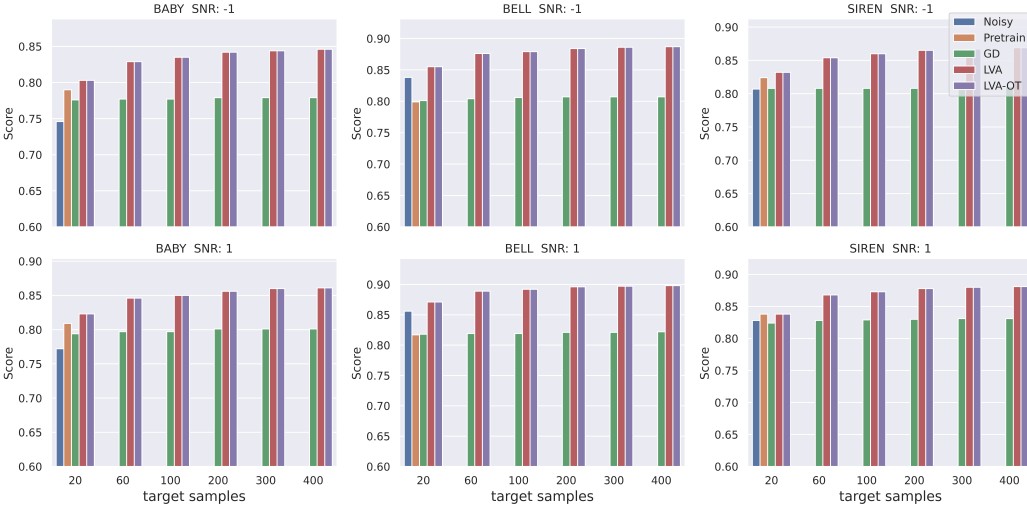

(b) STOI score of DDAE.

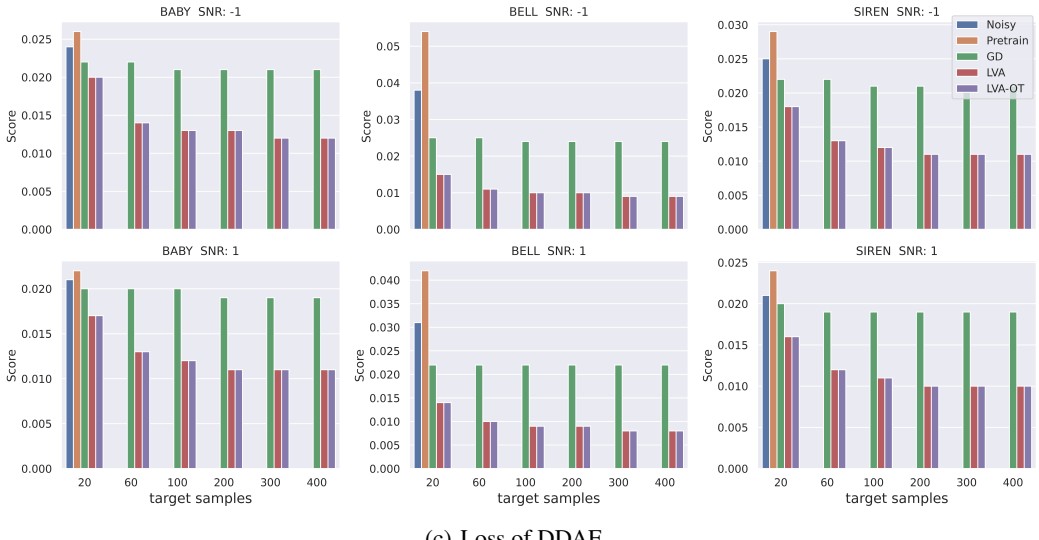

(c) Loss of DDAE.

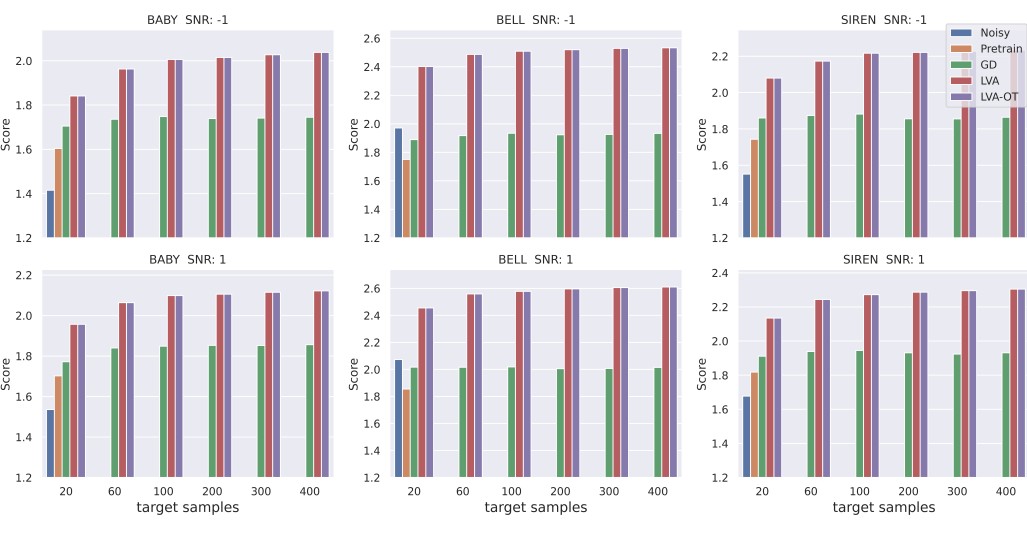

(d) PESQ score of BLSTM.

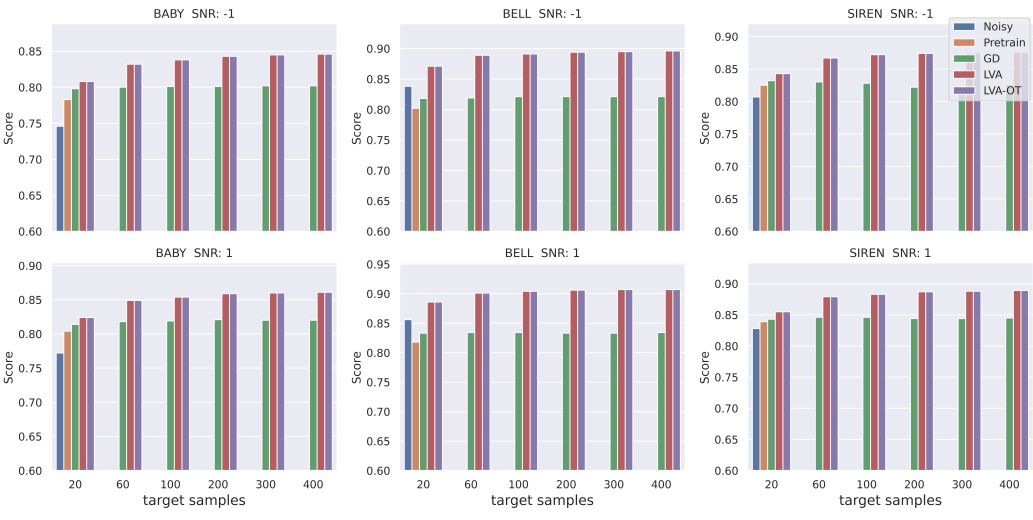

(e) STOI score of BLSTM.

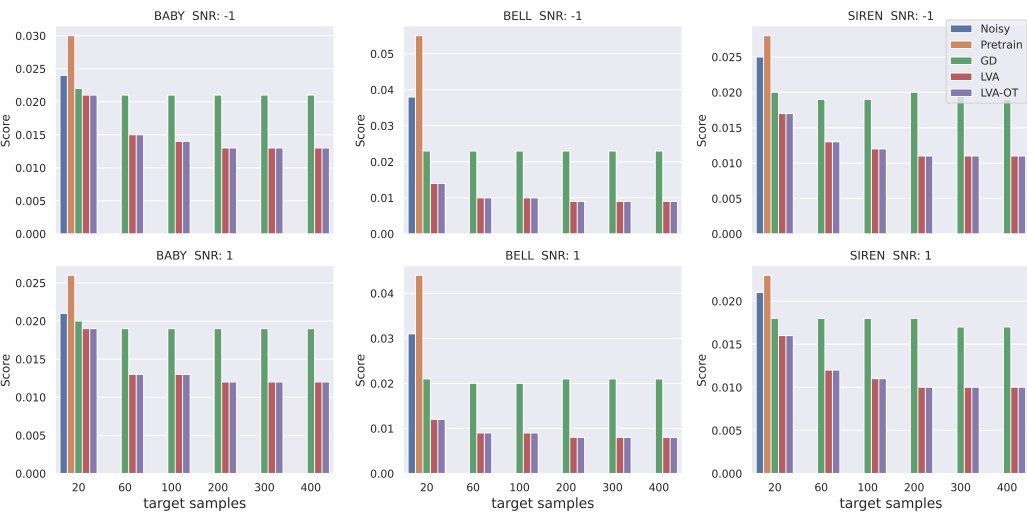

(f) Loss of BLSTM.

this section, we further investigate the achievable performance of LVA on SE where speakers are mismatched in the source and target domains.

First, we excluded the utterances of the speakers used in the training set and regenerated the adaptation and the test set by randomly sampling 200 utterances (100 for the adaptation set and 100 for the test set) from the remaining Deep Noise Suppression Challenge dataset. The experimental setup was same as Sec. 5.2 with testing speakers not given in the training set. We note that since the speakers did not match, an OT process was performed to align speakers. In Table 1, we denote LVA-OT as LVA for shorthand. The comparison of BLSTM model adaptation using GD and LVA is shown in Table 1. The table demonstrates that both GD and LVA improve the PESQ and STOI score after adaptation, while LVA consistently yields better performance over different noise types and SNR levels.

Table 1: Performance of finetuned models on mismatched speakers.

| noise type | SNR | noisy (no SE) | | Pretrain (no finetune) | | Finetuned by GD | | Finetuned by LVA | |
|---|---|---|---|---|---|---|---|---|---|
| | | PESQ | STOI | PESQ | STOI | PESQ | STOI | PESQ | STOI |
| Babycry | $-1$ | 1.40 | 0.74 | 1.58 | 0.78 | 1.58 | 0.79 | **1.88** | **0.81** |
| Babycry | $+1$ | 1.52 | 0.77 | 1.69 | 0.80 | 1.71 | 0.82 | **1.97** | **0.83** |
| Bell | $-1$ | 1.96 | 0.84 | 1.97 | 0.84 | 2.18 | 0.86 | **2.44** | **0.89** |
| Bell | $+1$ | 2.07 | 0.86 | 2.09 | 0.86 | 2.29 | 0.87 | **2.55** | **0.90** |
| Siren | $-1$ | 1.56 | 0.81 | 1.74 | 0.82 | 1.81 | 0.84 | **2.03** | **0.84** |
| Siren | $+1$ | 1.64 | 0.83 | 1.84 | 0.84 | 1.97 | 0.85 | **2.08** | **0.85** |
| Average | - | 1.69 | 0.81 | 1.82 | 0.82 | 1.92 | 0.84 | **2.16** | **0.85** |

### 8.2.2 ADAPTATION USING FULLSUBNET

In this extended experiment, we utilize a FullSubNet (Hao et al., 2021) pretrained network to perform SE domain adaptation tasks. The FullSubNet is a state-of-the-art SE model reaching the performance of PESQ: 2.89 and STOI: 0.96 on the DNS-challenge.

This implementation serves to examine the effect of a pretrained network. To compare with previous results of the BLSTM model in Sec. 8.2, same source domain data and the target domain set are used for adaptation. The FullSubNet receives an input of spectrum amplitude and outputs a complex Ideal Ratio Mask (cIRM) to compare with cIRM labels under $L_2$-loss. The architecture of the FullSubNet is composed of a full-band and a sub-band model. Within each model, two stacked LSTM layers and one fully connected layer are included. The LSTMs contain 512 hidden units and 384 hidden units in the full-band model and the sub-band model, respectively. The detailed structures can be found in (Hao et al., 2021).

A pretrained FullSubNet was obtained from the official Github[3], where the last linear layer was finetuned to adapt to the target set. The finetuned networks were subsequently evaluated on an adaptation test set to measure PESQ and STOI, where the results are shown in Table 2. The comparison shows that finetuning from a pretrained FullSubNet achieves enhanced baseline scores on the target domain (see Table 2 [no finetune]). While adaptation under the gradient descent significantly improves the pretrained network, the proposed LVA method outperforms the traditional gradient descent method in terms of PESQ and STOI. The results are consistent with the previous SE experiments to confirm that the proposed LVA promptly reaches better adaptation.

### 8.3 SUPER RESOLUTION FOR IMAGE DEBLURRING

**CNN extension** The LVA formulation Eq. (14) $\sim$ (19) can be extended to CNN layers with the following observation shown in Fig. 11, where the left-hand side depicts the usual convolution operation in CNNs. Given a CNN, denote the kernels by $\{\mathcal{C}_{ij\alpha\beta}\}$ with $(i, j, \alpha, \beta) \in L_1 \times L_2 \times C_{in} \times C_{out}$ and $L_1, L_2$ the kernel size (*e.g.,* $3 \times 3$), $C_{in}$ input channels and $C_{out}$ output channels. It is noticed

---

[3]https://github.com/haoxiangsnr/FullSubNet

Table 2: Performance of finetuned models from a pretrained FullSubNet.

| noise type | SNR | noisy (no SE) PESQ | noisy (no SE) STOI | Pretrain (no finetune) PESQ | Pretrain (no finetune) STOI | Finetuned by GD PESQ | Finetuned by GD STOI | Finetuned by LVA PESQ | Finetuned by LVA STOI |
|---|---|---|---|---|---|---|---|---|---|
| Babycry | $-1$ | 1.08 | 0.75 | 1.88 | **0.91** | 2.37 | 0.89 | **2.61** | **0.91** |
| Babycry | $+1$ | 1.09 | 0.77 | 2.05 | **0.92** | 2.63 | 0.91 | **2.75** | **0.92** |
| Bell | $-1$ | 1.06 | 0.84 | 2.10 | **0.94** | 2.92 | 0.93 | **3.00** | **0.94** |
| Bell | $+1$ | 1.08 | 0.86 | 2.29 | **0.95** | 3.05 | 0.94 | **3.11** | **0.95** |
| Siren | $-1$ | 1.17 | 0.81 | 2.58 | **0.94** | 2.92 | 0.93 | **3.02** | **0.94** |
| Siren | $+1$ | 1.18 | 0.83 | 2.80 | **0.95** | 3.06 | 0.94 | **3.16** | **0.95** |
| Average | - | 1.11 | 0.81 | 2.28 | **0.94** | 2.82 | 0.94 | **2.94** | **0.94** |

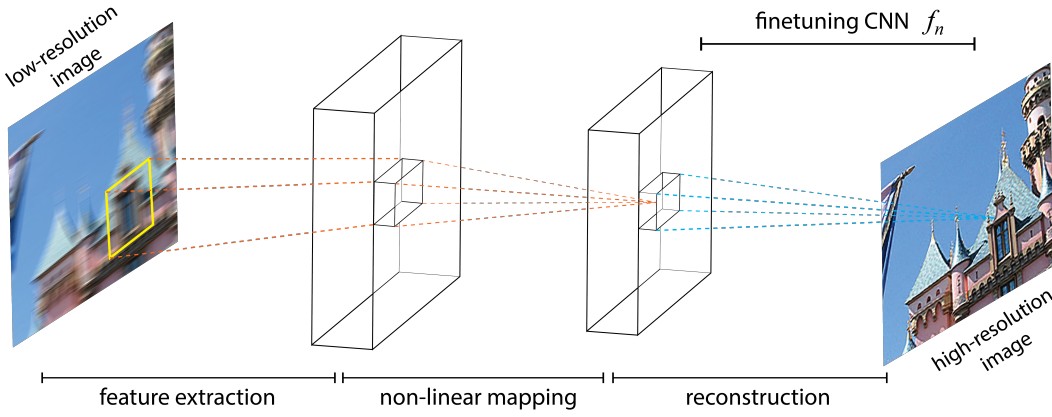

Figure 10: SRCNN for image deblurring. A pretrained model was composed by a sequence of CNN layers $f = f_n \circ \cdots f_2 \circ f_1$, where the last CNN layer $f_n$ was finetuned to fit more blurred images $\widetilde{\mathcal{D}}$.

that CNN kernels $\mathcal{C}_{ij\alpha\beta}$ cover a patch of the input image one at a time, denoted by $x_{ij\alpha}$; then each covered patch $x_{ij\alpha}$ has the same size as $\mathcal{C}_{ij\alpha\beta}$ for a fixed $\beta$. Consequently, each output pixel value at $(k, \ell, \beta)$ is computed by,

$$y_{\underline{k\ell}\beta} = \sum_{i,j,\alpha} \mathcal{C}_{\underline{ij\alpha}\beta} \cdot x_{\underline{ij\alpha}}(k, \ell) \tag{B.41}$$

where the underscore remarks indices are to be summed over. If we recall that a fully-connected layer of weights $\{W_{mn}\}$ calculates the output vector $\mathbf{v} = (v_1, \ldots, v_m, \ldots)$ from an input vector $\mathbf{u} = (u_1, \ldots, u_n, \ldots)$ by,

$$v_m = \sum_n W_{mn} \cdot u_n \tag{B.42}$$

then we noticed mathematically there is NO distinction between Eq. (B.41) and Eq. (B.42) as index $n$ has the role of $\underline{ij\alpha}$ and $m$ has the character of $\underline{k\ell\beta}$. In this view, CNN kernels $\mathcal{C}_{ij\alpha\beta}$ can be transformed to a 2D matrix $W_{mn}$ by an index conversion function $\eta$, and image patches $x_{ij\alpha}(k, \ell)$ can be transformed into a 1-D vector $u_n$ by another index function $\xi$, see Fig. 11 (RHS).

Thus, we completed the conversion of a CNN to a fully-connected-layer and naturally Eq. (17) $\sim$ (18) follow. There can also be possible extensions to other locally-connected networks in the future scope.

**Datasets** We used the CUration of Flickr Events Dataset (CUFED) (Wang et al., 2016) to train SRCNN (Scaman & Virmaux, 2018) for image deblurring, available at `https://acsweb.ucsd.edu/~yuw176/event-curation.html`. We randomly selected 2000 images from the CUFED and randomly crop 20 patches of size $33 \times 33$ from each image. Three pairs of training &

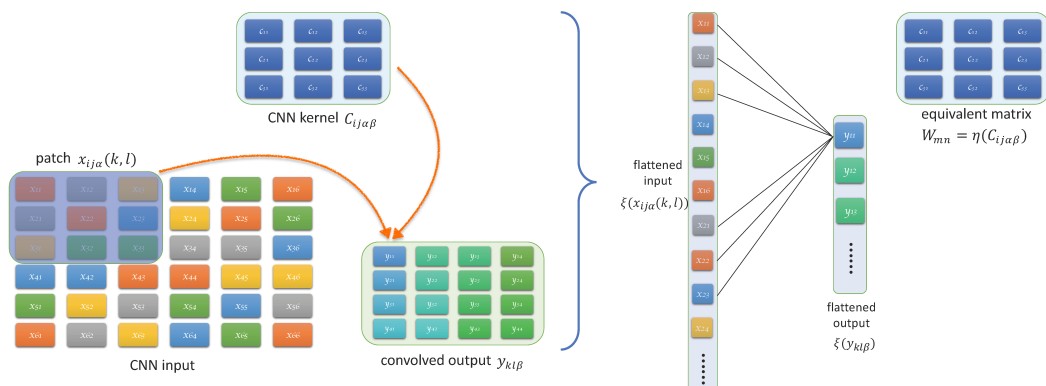

Figure 11: (LHS) The masked patch $x_{ij\alpha}$ is regarded as fully-connected under CNN kernels, Eq. (B.41). (RHS) By flattening the input and the convolved output into 1D vectors, a CNN is equivalent to a fully-connected layer Eq. (B.42).

finetune combination $\mathcal{D}$, $\widetilde{\mathcal{D}}$ were considered, see Fig. 12,

$$\mathcal{D}_1 = \left\{ (x_i, y_i) \mid y_i \in \{\text{original patches}\}, \, x_i \in \{\times 3 \text{ blurred patches}\} \right\}_{i=1}^{N=40K}$$
$$\widetilde{\mathcal{D}}_1 = \left\{ (\widetilde{x}_i, \widetilde{y}_i) \mid \widetilde{y}_i \in \{\text{original patches}\}, \, \widetilde{x}_i \in \{\times 6 \text{ blurred patches}\} \right\}_{i=1}^{N=40K}$$

(B.43)

$$\mathcal{D}_2 = \left\{ (x_i, y_i) \mid y_i \in \{\text{original patches}\}, \, x_i \in \{\times 3 \text{ blurred patches}\} \right\}_{i=1}^{N=40K}$$
$$\widetilde{\mathcal{D}}_2 = \left\{ (\widetilde{x}_i, \widetilde{y}_i) \mid \widetilde{y}_i \in \{\text{original patches}\}, \, \widetilde{x}_i \in \{\times 4, \times 6 \text{ blurred patches}\} \right\}_{i=1}^{N=40K}$$

(B.44)

and

$$\mathcal{D}_3 = \left\{ (x_i, y_i) \mid y_i \in \{\text{original patches}\}, \, x_i \in \{\times 3, \times 4 \text{ blurred patches}\} \right\}_{i=1}^{N=40K}$$
$$\widetilde{\mathcal{D}}_3 = \left\{ (\widetilde{x}_i, \widetilde{y}_i) \mid \widetilde{y}_i \in \{\text{original patches}\}, \, \widetilde{x}_i \in \{\times 6 \text{ blurred patches}\} \right\}_{i=1}^{N=40K}$$

(B.45)

Another image dataset SET14 (Zeyde et al., 2010) for testing finetuned models.

**Implementation**  The color images were first transformed to a different color space, YCbCr, to apply the SRCNN model on a single luminance channel. Our SRCNN, Fig. 10, receives $\mathcal{D}$ and $\widetilde{\mathcal{D}}$ of *low* resolution image patches to output *high* resolution ones of the same dimension. A pretrained model composed by a sequence of CNN layers $f = f_n \circ \cdots f_2 \circ f_1$ was pretrained by $\mathcal{D}$ under $L_2$-loss. Then the last CNN layer $f_n$ was finetuned to adapt to more blurred images $\widetilde{\mathcal{D}}$. Two finetuning methods by GD and LVA were compared. In particular, the conversion Eq. (B.41) $\Leftrightarrow$ Eq. (B.42) was used to admit LVA solutions.

Multiple pretrained architectures $f$ were implemented, of which were

$$
\begin{aligned}
f &= f_1 \circ f_2 \circ f_3, & &\text{(3-layer CNNs)} \\
f &= f_1 \circ f_2 \circ f_3 \circ f_4 \circ f_5, & &\text{(5-layer CNNs)} \\
f &= f_1 \circ f_2 \circ f_3 \circ f_4 \circ f_5 \circ f_6 \circ f_7 & &\text{(7-layer CNNs)}
\end{aligned}
$$

(B.46)

**Network Architecture**  We adopted the fast End-to-End Super Resolution model similar to the original architecture of SRCNN (Dong et al., 2016) with the insertion of a batch-normalization layer right after each ReLu activation function. The training batch size for $f$ was set at 128, and the testing batch size for gradient descent finetuning as 1.

The network architecture of pretrained models corresponding to Eq. (B.46):

1. 3-layer-CNN of kernel size: $(9, 5, 5)$ with channel size: $(1, 32, 32, 1)$.

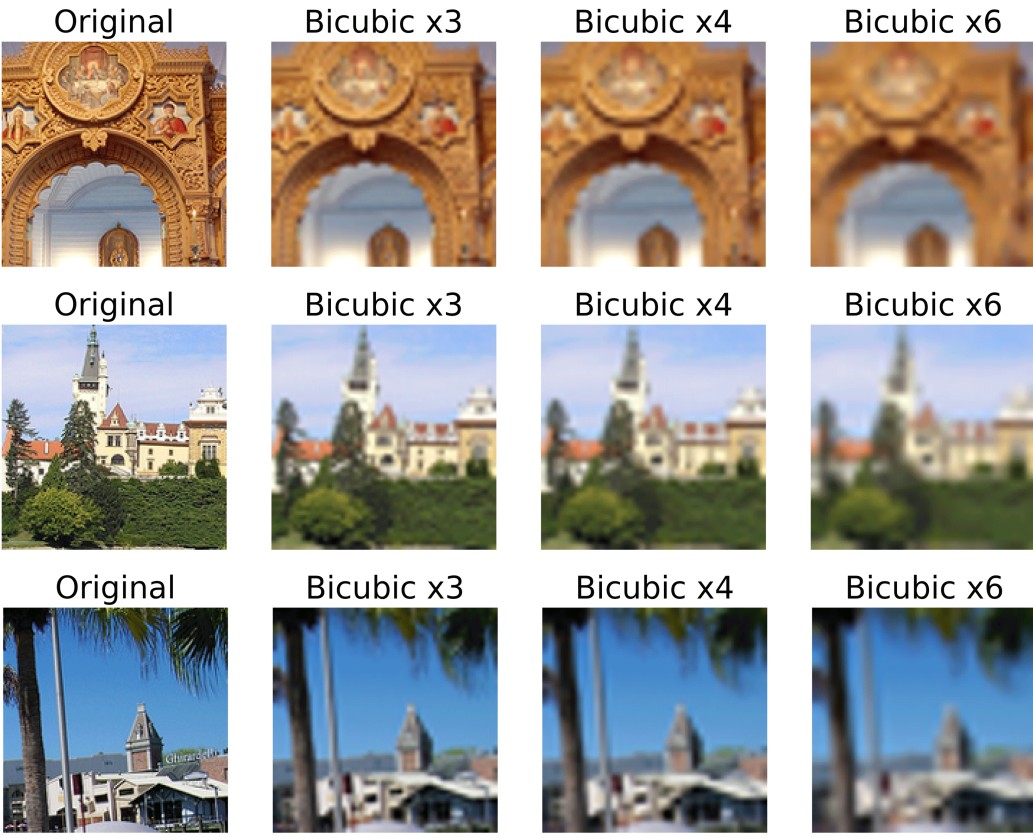

Figure 12: Samples of CUFED for training and finetuning (bicubic$\{\times 3, \times 4, \times 6\}$) for Eq. (B.43)$\sim$(B.43).

2. 5-layer-CNN of kernel size: $(9, 5, 5, 5, 5)$ channel size: $(1, 32, 32, 32, 32, 1)$.

3. 7-layer-CNN of kernel size: $(9, 5, 5, 5, 5, 5, 5)$, channel size: $(1, 32, 32, 32, 32, 32, 32, 1)$.

where the first and the last number in the channel size is always 1 to indicate the input and output luminance channel.

**More extensive results**  Fig. 13 shows the deblurring results by finetuned models of GD and the LVA, where our method reached the highest PSNR with the least finetuning samples $N = 256$. This outcome indicated the LVA method is beneficial for few-shot learning, especially when adaptation data is few.

More extensive experiments were conducted to observe performance. The following results compared the finetune loss and PSNR with different combinations of training & finetune data in Eq. (B.43)$\sim$(B.45) and multiple pretrained architectures of $f$ in Eq. B.46.

In Fig. (a) $\sim$ (f), the horizontal axis of each subfigure is the patch number and the title of each subfigure indicated the transition from the old data $\mathcal{D}$ to new data $\widetilde{\mathcal{D}}$ by $\mathcal{D} \to \widetilde{\mathcal{D}}$, *e.g.,* $\{\times 3, \times 4\} \to \times 6$.

Overall, the following facts were observed in Fig. (a)$\sim$(f):

1. The loss is roughly inverse-proportional to PSNR score. Increasing CNN layers can handle more difficult tasks, such as $\{\times 3\} \to \{\times 4, \times 6\}$ compared to $\{\times 3\} \to \{\times 6\}$ and $\{\times 4, \times 6\} \to \{\times 3\}$ since $\{\times 3\} \to \{\times 4, \times 6\}$ was asked to adapt to more new data than it was originally trained.

2. In all figures, the LVA method had stable performance over GD in both loss and PSNR even when sample sizes ($x$-axis) largely decreased. We can see ups and downs in orange bars, but not blue bars.

3. Obviously in Fig. (a), GD showed difficulty in learning $\{\times 3\} \to \{\times 4, \times 6\}$ and $\{\times 3\} \to \{\times 6\}$ than $\{\times 3, \times 4\} \to \{\times 6\}$, but the LVA method remained roughly the same (stable) as well as retaining its best performance over GD.

4. Overall, GD method improvement was not consistent with the data number, likely due to the local minimum occurring in neural networks. Conversely in the LVA method, the performance is (stably) proportional to data samples. Therefore, one could easily derive acceptable results with few finetune data.

Fig. (a) $\sim$ (f) then concludes that the LVA method has better performance and is more efficient than GD in terms of adaptation data amount considered.

**Hardware**  One NVIDIA V100 GPU (32 GB GPU memory) with 4 CPUs (128 GB CPU memory).

**Runtime**  Approximately 4 minutes in the LVA method and 15 minutes in the GD method with 256 patches (1,000 epochs).

## 8.4  Additional Experiments on Domain Adaptive Image Classifications

We conduct two additional experiments on images to investigate domain adaptation ability using (1) Office-31 (Saenko et al., 2010) and (2) MNIST $\to$ USPS.

**Office-31:**  Table 3 shows the results of domain transitions within the 3 domains of Office-31: *Amazon (A), DSLR (D) and Webcam (W)*, resulting in 6 possible transitions to classify 31 items. On each individual source domain ($A$, $D$, $W$), the pretrained model was trained for 30 epochs. Subsequently, it was finetuned onto target domains for another 30 epochs. The proposed LVA obtained higher adaptation accuracy in most domain transitions, confirming that LVA can promptly adapt to new tasks.

Table 3: Office-31: adaptation accuracy between 3 domains.

| **Domain** | A$\to$D | A$\to$W | D$\to$A | D$\to$W | W$\to$A | W$\to$D |
|---|---|---|---|---|---|---|
| pretrain | 77.4 % | 67.1 % | 54.2 % | 80.7 % | 59.4 % | 92.9 % |
| GD | 78.7 % | 71.0 % | 58.7 % | 86.5 % | 60.6 % | **94.2 %** |
| LVA | **84.5 %** | **79.4 %** | **71.0 %** | **87.7 %** | **70.3 %** | 93.6 % |

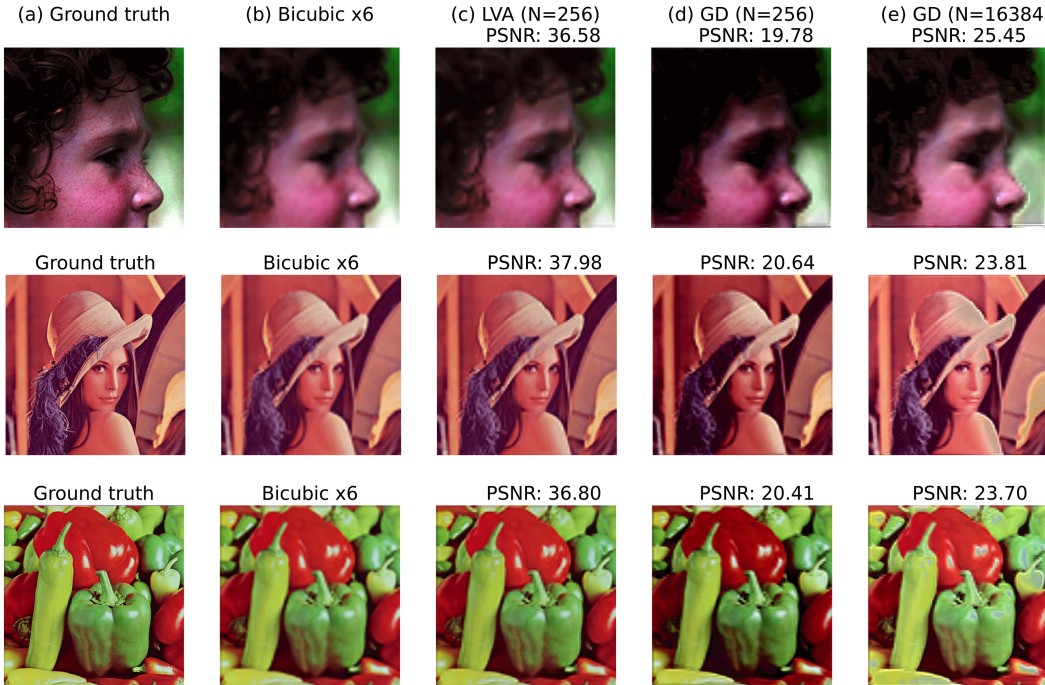

Figure 13: Results of image deblurring by SRCNN. Column (b): testing input images from SET14; column (c): results of LVA model finetuned with 256 patches; column (d): results of GD model finetuned with 256 patches; column (e): results of GD model finetuned with 16,384 patches. The corresponding PSNR shown in the figures indicated our LVA method reached the best PSNR with the least finetuning samples.

**MNIST → USPS:** In this experiment, the pretrained model was trained on MNIST (source domain) with 98.94% accuracy by 20 epochs. While adapting to the target domain USPS, the accuracy of the pretrained model dropped rapidly to 65.32%. Subsequently, gradient descent (GD) and the proposed LVA were deployed to enhance the target domain results. Table 4 shows the adaptation accuracy attained at different finetune epochs. It was observed that LVA consistently outperformed GD and reached high accuracy in just a few epochs. These two additional experiments on image classifications again verify that the proposed LVA is capable of real-world DA tasks; the adaptability is general.

Table 4: MNIST → USPS (classification accuracy)

| Finetune method | 10 epochs | 50 epochs | 90 epochs | 120 epochs |
|---|---|---|---|---|
| pretrain | 65.32% | | | |
| GD | 74.64% | 80.12% | 82.06% | 82.46% |
| LVA | **86.90**% | **90.13**% | **90.83**% | **90.98**% |

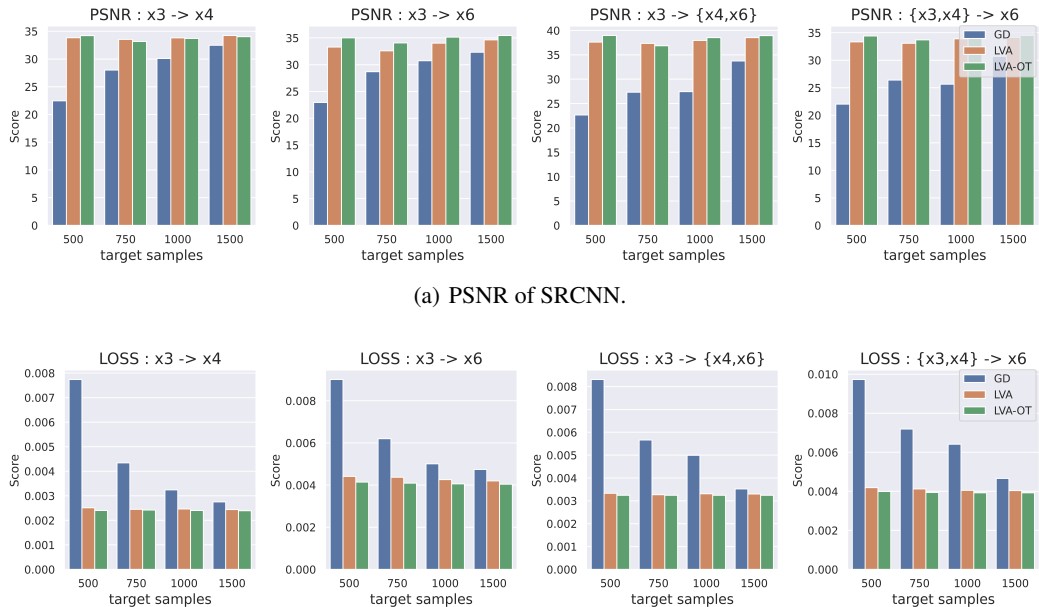

(a) PSNR of SRCNN.

(b) Loss of SRCNN.

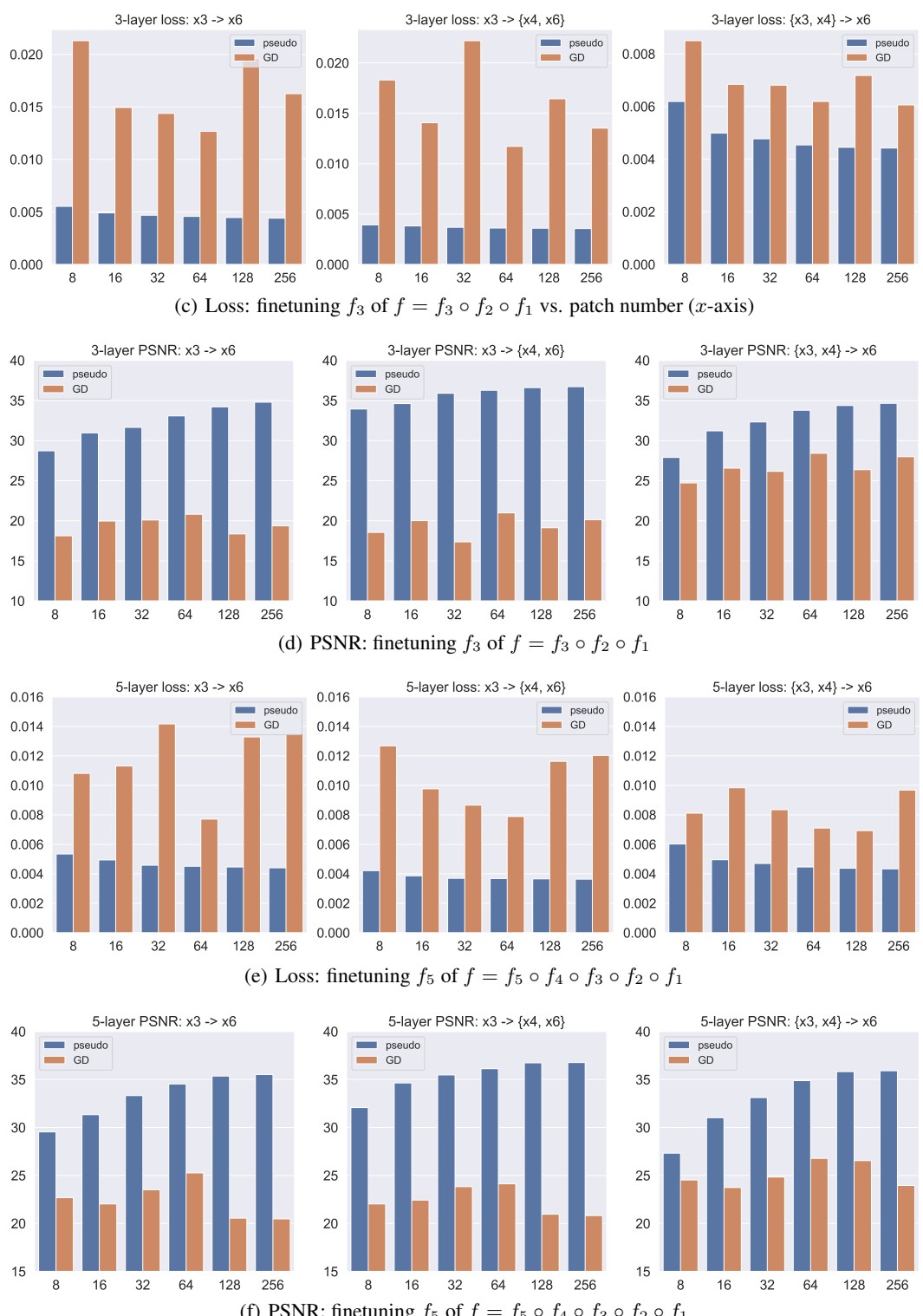

(c) Loss: finetuning $f_3$ of $f = f_3 \circ f_2 \circ f_1$ vs. patch number ($x$-axis)

(d) PSNR: finetuning $f_3$ of $f = f_3 \circ f_2 \circ f_1$

(e) Loss: finetuning $f_5$ of $f = f_5 \circ f_4 \circ f_3 \circ f_2 \circ f_1$

(f) PSNR: finetuning $f_5$ of $f = f_5 \circ f_4 \circ f_3 \circ f_2 \circ f_1$

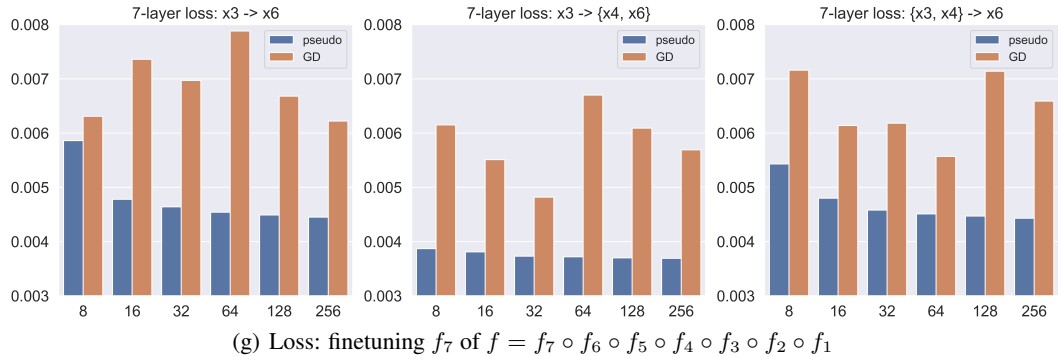

(g) Loss: finetuning $f_7$ of $f = f_7 \circ f_6 \circ f_5 \circ f_4 \circ f_3 \circ f_2 \circ f_1$

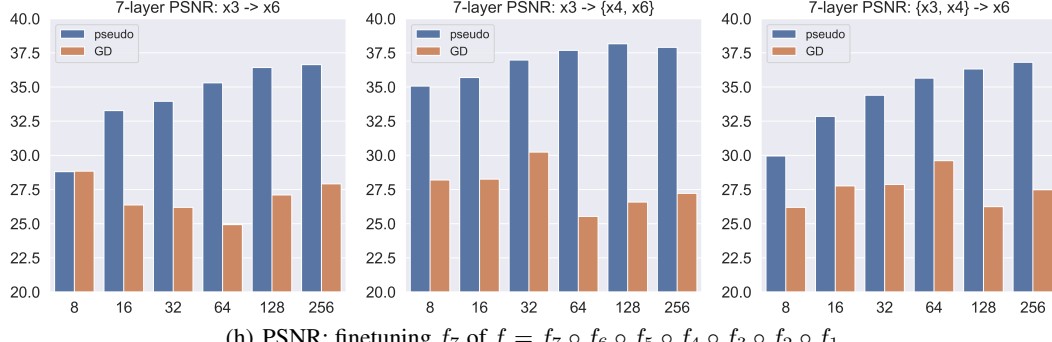

(h) PSNR: finetuning $f_7$ of $f = f_7 \circ f_6 \circ f_5 \circ f_4 \circ f_3 \circ f_2 \circ f_1$

