# OpenReview forum: "Interpretations of Domain Adaptations via Layer Variational Analysis"
_ICLR.cc/2023/Conference — ICLR 2023 poster_

### Official Review · Reviewer_2KJc · 2022-10-21

**Confidence:** 3
**Correctness:** 2
**Technical Novelty And Significance:** 3
**Empirical Novelty And Significance:** 3
**Recommendation:** 6

**Clarity, Quality, Novelty And Reproducibility:**

Clarity: most part of the paper is clear except for those mentioned in questions above.

Quality: the overall quality is good, with some concerns on the theory and experiments as discussed above.

Novelty: the proposed method is novel to my knowledge.

Reproducibility: code is available.

**Strength And Weaknesses:**

**Strengths**

- The paper has a good overview of the literature.
- The method of layer-wise optimization (from the $\delta f_n$ point of view) is novel. Eq (17)-(18) gives new insights to transfer learning.
- Experiments are conducted on multiple domains with consistent improvement over the GD baseline.

**Weaknesses**

- The paper does not explicitly define norms such as $\\| \cdot \\|_{\\mathcal{Y}}$.
- The data deviation assumption might be strong in practice.
- Thm 3.6 is not clear and does not provide much insight. Do you mean $\exists \delta_f$ or $\forall \delta f$? Does it need to satisfy the constraint that $f+\delta f$ is a network layer? Also, simply by setting $\delta f==0$ should yield eq (10); in that case, why do you have this theorem.
- Eq (16): it seems to be $\approx$ instead of $=$ because $f_n$ is an affine transformation followed by activation. This might also introduce further error terms in the following equations, so I would like to see analysis on that. E.g., a higher-order analysis.
- Around eq (21): it should be $\min$ eq (17) instead of (19). That $\delta f_n$ is linear seems infeasible due to non-linear activation, so I think mentioning the pseudo-inverse is unnecessary.
- Experiments on speech enhancement:
  - Do you use mismatched speakers of clean speech between training and adaption sets?
  - Why is the fmax=6k given that training data are 16kHz?
  - It would be interesting to test on the no-reverb test set as it provides different noise types and part of them can be used as the adaptation set.
  - The baseline is sort of weak. More recent denoisers can achieve test pesq > 2.8 or even 3.0 (see e.g. FullSubNet, Demucs, CleanUNet, ...). I think it's feasible to take the pretrained models (all of these models have) and conduct adaptation to new noise types and languages in DNS-2021/2022. If it comes out with good results then the paper can have more impact.
- Experiments on image super-resolution/de-blurring: the major concern is that there is no quantitative results, so it is hard to evaluate the methods fairly.

**Summary Of The Paper:**

The paper proposes a new aspect to look at transfer learning. The basic idea is to allow each layer $f_n$ to change a little bit ($\delta f_n$). This gives bounds on losses on the new dataset ($L(g)$) under a few assumptions. Based on this framework, $L(g)$ can be re-written as a least-square problem, and $\delta f_n$ can be optimized for the new dataset. The paper conducts experiments on a synthetic dataset, a speech enhancement task, and image de-blurring/super-resolution. Results show the proposed method outperforms GD.

**Summary Of The Review:**

Based on the questions and concerns above, I think the paper has some problems in theory and space for improvement in experiments. Therefore I think this version cannot be accepted. I will make the final decision based on authors' response and revised version.

----------

After rebuttal:

The authors improved the experiments and I found them good, so I'll increase my score to 6. I think the higher-order analysis can be further improved, especially for more standard neural networks with Lipschitz activations.

---

> ### Author Response · Authors · 2022-11-20
> **Response 3 to Reviewer 2KJc**
>
> > **6.3 [SE experiments] It would be interesting to test on the no-reverb test set as it provides different noise types and part of them can be used as the adaptation set.**
>
> The main focus of this study is model adaptation. So we need to prepare an adaptation set and a testing set. However, the "no-reverb test set" only contains a testing set, and thus we could not directly use the "no-reverb test set" to test our system. Therefore, we used speech utterances from the DNS dataset to mix with our own noise types and specific SNR levels to generate the adaptation and testing sets.
>
> In addition, for the "no-reverb test set", there are no explicit SNR labels available. In this study, we intended to systemically investigate the adaptation performance under different scenarios, including noise type and SNR levels. Thus, we have designed our own adaptation and testing sets, where the effects of SNR levels can be explicitly investigated.
>
> > **6.4 [SE experiments] The baseline is sort of weak. More recent denoisers can achieve test $\text{PESQ} > 2.8$ or even 3.0 (see e.g. FullSubNet, Demucs, CleanUNet, ...). I think it's feasible to take the pretrained models (all of these models have) and conduct adaptation to new noise types and languages in DNS-2021/2022. If it comes out with good results then the paper can have more impact.**
>
> To eliminate the concern of weak baseline models, on the reviewer's suggestion we replace the baseline model with a pretrained FullSubNet. By using an officially released FullSubNet as our pretrained model, we arrived at PESQ: 2.89 and STOI: 0.96 on the DNS challenge as asserted. The stronger pretrain subsequently resulted in enhanced performances as shown in the following table. It can be noticed that the *pretrained model without finetune* alone can achieve high performance of PESQ: 2.80 and STOI: 0.95 (in Siren, SNR=+1) on our test set.
>
> While the baseline scores are high, finetuning by GD and LVA still gains much improvement in PESQ. Notably, finetuning by LVA remains outperforming the traditional gradient descent method. More detailed experimental settings are described in the revision Section 8.1 (Appendix).
>
> - **Table 2:** Performance of finetuned models from a pretrained FullSubNet
> |    |   |  <noisy | (no SE)> | <Pretrain | (no fine-tune)> | <Finetune | by GD> |  <Finetune | by LVA> |
> | ---------------  |:------:| :------:| :------:| :------:| :------:|:------:| :------:|:------:| :------:|
> | **noise type**  | SNR  | PESQ | STOI | PESQ | STOI | PESQ | STOI | PESQ | STOI |
> | Babycry  | -1 | 1.08 | 0.75 | 1.88 | **0.91** | 2.37 | 0.89 | **2.61** | **0.91** |
> | Babycry  | +1 | 1.09 | 0.77 | 2.05 | **0.92** | 2.63 | 0.91 | **2.75** | **0.92** |
> | Bell     | -1 | 1.06 | 0.84 | 2.10 | **0.94** | 2.92 | 0.93 | **3.00** | **0.94** |
> | Bell     | +1 | 1.08 | 0.86 | 2.29 | **0.95** | 3.05 | 0.94 | **3.11** | **0.95** |
> | Siren    | -1 | 1.17 | 0.81 | 2.58 | **0.94** | 2.92 | 0.93 | **3.02** | **0.94** |
> | Siren    | +1 | 1.18 | 0.83 | 2.80 | **0.95** | 3.06 | 0.94 | **3.16** | **0.95** |
> | Average  | -  | 1.11 | 0.81 | 2.28 | **0.94** | 2.82 | 0.94 | **2.94** | **0.94** |
>
> > **7. Experiments on image super-resolution/de-blurring: the major concern is that there is no quantitative results, so it is hard to evaluate the methods fairly.**
>
> In the super-resolution/de-blurring task, the PSNR index is used as an objective measurement to indicate the improvement of image deblur, which shares the same evaluation approach as the paper: [**Image super-resolution using deep convolutional networks**](https://arxiv.org/pdf/1501.00092.pdf)
>
> In general, higher PSNR indicates higher reconstruction image quality.
>
> > **[Summary Of The Review]
> Based on the questions and concerns above, I think the paper has some problems in theory and space for improvement in experiments. Therefore \textbf{I think this version cannot be accepted. I will make the final decision based on authors' response and revised version.**
>
> The reviewer's detailed comments and helpful suggestions are deeply appreciated to guide us to improve.

---

> ### Author Response · Authors · 2022-11-20
> **Response2 to Reviewer 2KJc**
>
> > **4. Eq (16): it seems to be $\simeq$ instead of $=$ because $f_n$ is an affine transformation followed by activation. This might also introduce further error terms in the following equations, so I would like to see analysis on that. E.g., a higher-order analysis.**
>
> We thank the reviewer for bringing up the careful question. In carrying out Eq. (16), our first intention was to study a regression finetune problem by considering $L_2$-loss and a linear layer $f_n$ with only identity activation function ($\sigma_n = I$). Therefore, if $f_n$ is purely an affine function $f_n (z) = A_n(z) = W_n \cdot z + b$, previous equation $f_n(\widetilde{z}_i) - f_n(z_i) = W_n \cdot (\widetilde{z}_i - z_i) = J(f_n)(z_i) \cdot \delta z_i $ is an exact identity. However, owing to the valuable comment, we understand it is also important to show higher-order analysis so that the whole complexity can be seen. Therefore, higher-order corrections of Eq. (16) are now **shown in the revision for completion**. In the end, we take the advantage of the linear $f_n$ to derive an exact solution, which demonstrates an interesting case of LVA. An amendment was made in the revised manuscript after Eq. (16) to notify readers of the reviewer's concern and our simplifications.
>
> > **5. Around eq (21): it should be $\min$ eq (17) instead of (19). That $\delta f_n$ is linear seems infeasible due to non-linear activation, so I think mentioning the pseudo-inverse is unnecessary.**
>
> We thank the Review for pointing out the typo in minimizing Eq.(17) instead of Eq.(19), which is now **corrected in the revision**.
>
> As addressed by the reviewer, $\delta f_n$ can indeed include non-trivial activations as in Eq.~(16). However, the purpose of examining the linear last-layer case is *two-fold*: (1) theoretical computations can be done **explicitly** to analyze empirical transfer learning techniques without always referring to the network as a black box, and (2) the case corresponds to regression problems directly, where our formulation shows good results in applications (the Experiment Section) and *meaningful interpretations* for transfer learning (see paragraph **Knowledge Transfer**).
>
> In such case, we observe that finetuned networks are equipped with **self-correction ability** using pretrain knowledge $x_i$ and $J(f)$ in Eq. (20) to **annihilate new domain label change**. We believe this property is important and fundamental to justify the term "transfer leaning" because previous domain knowledge is indeed carried over for adaptation. To our understanding, these highlights of our study are not found in the existing literature. Therefore, we consider linear $\delta f_n$ still carry theoretical significance to be reported and studied.
>
>
> > **6.1 [SE experiments] Do you use mismatched speakers of clean speech between training and adaption sets?**
>
> No, our speakers coincide in the training set and the adaptation set with differences in noise types and SNRs. Such design was fixed across three models (pretrained net, GD-finetuned net, and LVA-finetuned net) to compare in the SE experiment, and therefore this was posed as a fair comparison.
>
> However, it can be understood that the reviewer may have concerns regarding the matched speakers, and therefore we have repeated the experiment using a mismatched speaker test set. The results are shown below and as well as amended in the revision Section 8.2.1 (Appendix) for readers' reference.
>
> - **Table 1:** Performance of finetuned models toward mismatched speakers.
> |    |   |  <noisy | (no SE)> | <Pretrain | (no fine-tune)> | <Finetune | by GD> |  <Finetune | by LVA> |
> | ---------------  |:------:| :------:| :------:| :------:| :------:|:------:| :------:|:------:| :------:|
> | **noise type**  | SNR  | PESQ | STOI | PESQ | STOI | PESQ | STOI | PESQ | STOI |
> | Babycry  | -1 | 1.40 | 0.74 | 1.58 | 0.78 | 1.58 | 0.79 | **1.88** | **0.81** |
> | Babycry  | +1 | 1.52 | 0.77 | 1.69 | 0.80 | 1.71 | 0.82 | **1.97** | **0.83** |
> | Bell  | -1 | 1.96 | 0.84 | 1.97 | 0.84 | 2.18 | 0.86 | **2.44** | **0.89** |
> | Bell  | +1 | 2.07 | 0.86 | 2.09 | 0.86 | 2.29 | 0.87 | **2.55** | **0.90** |
> | Siren  | -1 | 1.56 | 0.81 | 1.74 | 0.82 | 1.81 | 0.84 | **2.03** | **0.84** |
> | Siren  | +1 | 1.64 | 0.83 | 1.84 | 0.84 | 1.97 | 0.85 | **2.08** | **0.85** |
> | Average  | -  | 1.69 | 0.81 | 1.82 | 0.82 | 1.92 | 0.84 | **2.16** | **0.85** |
>
>
> > **6.2 [SE experiments] Why is the fmax=6k given that training data are 16kHz?**
>
> Indeed, 16kHz wave files originally rendered spectrograms of 8kHz frequency bins. However, due to that higher frequencies did not yield much deviation, we only show major differences below 6kHz for visualization.

---

> ### Author Response · Authors · 2022-11-20
> **Response 1 to Reviewer 2KJc**
>
>
> >  **1. The paper does not explicitly define norms such as $|| \cdot ||\_{\mathcal{Y}}$.**
>
> As this study intends to keep our framework as general as possible, one finds that a general norm $|| \cdot ||\_{\mathcal{Y}}$ holds true throughout most of the computation of Sec. 3 & 4, except for Sec 4.2 where the last Moore-Penrose pseudo-inverse is computed under the Euclidean norm, where we emphasized: "**... if $\delta f_n$ is a linear functional with $|| \cdot ||\_{\mathcal{Y}}$ being $L_2$-norm**" in line before Eq. (21) to remind readers.
>
> In our experiments, $|| \cdot ||\_{\mathcal{Y}}$ is set to be $L_2$-norm in practice, where we have specified the use of $L_2$-norm **in the revised manuscript, Section 8** (Appendix): Experimental Details.
>
>
> > **2. The data deviation assumption might be strong in practice.**
>
> To be able to provide a theory discussing finetuning mechanism and the relation between the loss bounds, a definition characterizing the domain difference is necessary.
>
> As such, we decided to write the definition of data deviation in a stronger version so that Theorem 3.6 can be derived succinctly, and the proof can be clear and easy to grasp. In fact, a more delicate form can be devised to include outliers in domains for computing adaptation loss, only that cumbersome analysis quickly follows.
>
> Another aspect to alleviate the reviewer's concern is that: our proposed method can be applied to any two largely deviating domains with a chosen threshold for data deviation in alignment. A sample with a larger deviation may be excluded as an outlier, and the threshold can be considered as a hyper-parameter in training.
>
> Aside from invoking domain alignment, it has been noticed that in practice our experiments in speech enhancement and image deblurring tasks have verified that the LVA is well applied to large datasets. Recent additional experiments on image classification task **Office-31** have been **added in Section 8.4 (Appendix)** to further support that our proposed method is well-suited for real-world domain adaptations.
>
> > **3. Thm 3.6 is not clear and does not provide much insight. Do you mean $\exists \delta f$ or $\forall \delta f$ ? Does it need to satisfy the constraint that $f + \delta f$ is a network layer? Also, simply by setting $\delta f ==0$ should yield eq (10); in that case, why do you have this theorem.**
>
> Theorem 3.6 describes the relationship between domain difference and the corresponding finetune loss. Consequently, it explains why the fixed-layered technique can be empirically successful for transfer learning. This is (**to reviewer's question**) why **we have Theorem 3.6 first** as foundations.
>
> Theorem 3.6 uniquely views the **network layer change** as **functional variations** $\delta F$ on *pretrained layers* $f = F \circ F_{n-r}$ to derive that any finetuned network $g= (F + \delta F) \circ F_{n-r}$ has a bounded loss, subject to the model and the task applied. Therefore, (**to reviewer's question**) it is $\forall \delta F$ that Theorem 3.6 holds as long as $\delta F$ is Lipschitz. This is (**to reviewer's question**) particularly true when $F + \delta F$ satisfies the network layer condition, a special case.
>
> Despite that Theorem 3.6 states that a finetune $g= (F + \delta F) \circ F_{n-r}$ (with Lipschitz $\delta F$) naturally associates with a bounded transfer loss, we want to know: **what variation $\delta F$ minimizes the target (domain) loss?** The **insight (to reviewer's question)** was in fact revealed by the derivation of Theorem 3.6 to lead to the optimal $\delta F$ in **Sec. 4**, namely our LVA: Eq. (17)$\sim$(20).
>
> As $\delta F$ stands for the variation to adapt to a new task, it is rarely $\delta F \equiv 0$. In fact, (to reviewer's question) $\delta F \equiv 0$ automatically implies $g = (F + \delta F) \circ F_{n-r} = F \circ F_{n-r} =f$, so that the finetune $g$ has to coincide with the pretrain $f$ at all times, which is unlikely a desirable scenario. As such, the optimal variation $\delta F \neq 0$ in general and in the 1-layer finetuning case the solution is given by Eq. (17)$\sim$(20).

---

### Official Review · Reviewer_xrcZ · 2022-10-24

**Confidence:** 2
**Correctness:** 3
**Technical Novelty And Significance:** 3
**Empirical Novelty And Significance:** 3
**Recommendation:** 6

**Clarity, Quality, Novelty And Reproducibility:**

The proposed framework allows us to finetune the model by only adding a residual but the residual can be different to compute and may not even be applicable for many cases.

**Strength And Weaknesses:**

pros:

1. It is interesting to see we can only add a Lipschitz function $\delta_f$ to finetune on different datasets.


Cons:

1. The framework requires sample alignment first. To me, the sample alignment can be a very huge cost and may not be applicable in most finetuning cases. For example,  given a pre-trained ImageNet classifier and a new dataset such as Office31, we have to align samples first. It can be challenging to measure the difference and thus align successfully. If we can align them, we may just use the aligned image and labels to train the model. So we don't need to finetune.

2. The experiment results are not enough and I would like to see quantitative comparisons.  In particular, since the paper title suggests the domain adaptation, I would like to see how can we benefit from the proposed framework on popular domain adaptation datasets, e.g., DomainNet, Office31.

**Summary Of The Paper:**

The paper proposes a theory framework to enable better finetuning by only adding a Lipschitz function.

**Summary Of The Review:**

The proposed framework allows us to finetune the model by only adding a residual but the residual can be different to compute and may not even be applicable for many cases.

---

> ### Author Response · Authors · 2022-11-20
> **Response to Reviewer xrcZ**
>
> >  **1. [Weaknesses] The framework requires sample alignment first. To me, the sample alignment can be a very huge cost and may not be applicable in most finetuning cases. For example, given a pre-trained ImageNet classifier and a new dataset such as Office31, we have to align samples first. It can be challenging to measure the difference and thus align successfully. If we can align them, we may just use the aligned image and labels to train the model. So we don't need to fine-tune.**
>
> We thank the reviewer for raising this important point from a practical perspective. Indeed, alignments need to be operated efficiently, especially in real-world scenarios. The computational expense can be controlled based on efficient sorting algorithms and potentially selected alignments. Specifically, we can allow only partial alignments on the samples whose distances with the source domain exceed a certain threshold. The threshold can be chosen depending on the dataset and available resources.
>
> Furthermore, our framework admits 1-step solving process to obtain prompt adaptation, which is efficient in time and requires fewer resources than direct gradient descent training using aligned images and labels.
>
> >  **2. The experiment results are not enough and I would like to see quantitative comparisons. In particular, since the paper title suggests the domain adaptation, I would like to see how can we benefit from the proposed framework on popular domain adaptation datasets, e.g., DomainNet, Office31.**
>
> We thank the reviewer for the helpful comments. On the reviewer's suggestion, additional quantitative comparisons were conducted accordingly, where two popular domain adaptation tasks **Office-31** & **MNIST** $\to$ **USPS** were demonstrated as follows.
>
> **Table 1** shows the results of Office-31, where the domain transitions occurred within the 3 domains of Office-31: *Amazon (A)*, *DSLR (D)* and *Webcam (W)*, resulting in 6 possible transitions to classify 31 items. On each individual source domain (A, D, W), the pretrained model was trained for 30 epochs. Subsequently, it was finetuned onto target domains for another 30 epochs. The proposed LVA obtained higher adaptation accuracy in most domain transitions, confirming that LVA can promptly adapt to new tasks.
>
> - **Table 1:** Office-31: adaptation accuracy between 3 domains.
> | Domain | A $\to$ D |  A $\to$ W | D $\to$ A | D $\to$ W |  W $\to$ A | W $\to$ D |
> | :---------|:-----------:| :-----------:| :-----------:| :-----------:| :-----------:| :-----------:|
> | pretrain  | 77.4%  | 67.1% | 54.2% | 80.7% | 59.4% | 92.9% |
> | GD         | 78.7%  | 71.0% | 58.7% | 86.6% | 60.6% | **94.2%** |
> | LVA        |  **84.5%** | **79.4%** |  **71.0%** | **87.7%** | **70.3%** | 93.6% |
>
> **Table 2** shows the resulting accuracy under adaptation: **MNIST** $\to$ **USPS**:
>
> - **Table 2**: MNIST $\to$ USPS (classification accuracy)
> | Finetune method | 10 epochs |  50 epochs | 90 epochs | 120 epochs |
> | :-------------------|:------------:| :------------:| :------------:| :-------------:|
> | pretrain  | 65.32%   |  65.32%   | 65.32% |  65.32%   |
> | GD         | 74.64%  | 80.12% | 82.06%  | 82.46% |
> | LVA        |  **86.90%** | **90.13%** |  **90.83%** | **90.98%** |
>
> In this experiment, the pretrained model was trained on MNIST (source domain) with 98.94% accuracy by 20 epochs. While adapting to the target domain USPS, the accuracy of the pretrained model dropped rapidly to 65.32%. Subsequently, gradient descent (GD) and the proposed LVA were deployed to enhance the target domain results. Table 1 provides the accuracy attained at different finetune epochs. It was observed that LVA consistently outperformed GD and reached high accuracy in just a few epochs.
>
> These two additional experiments on image classifications again verify that the proposed LVA is capable of real-world DA tasks; the adaptability is general.
>
> Due to this useful suggestion by the reviewer, the above experiments and results have been **amended to Appendix 8.4 (supplemental materials) and promptly mentioned in Sec. 5**, owing to the limited space allowed in the main article. The codes and the corresponding results will also be released on GitHub soon after the organization.

---

> > ### Comment · Reviewer_xrcZ · 2022-12-09
> > **Thanks for your response**
> >
> > I have read the response and my concern about the dataset is addressed. But I'm still not fully convinced about the sample alignment time.
> >
> >  **The computational expense can be controlled based on efficient sorting algorithms and potentially selected alignments. Specifically, we can allow only partial alignments on the samples whose distances with the source domain exceed a certain threshold. The threshold can be chosen depending on the dataset and available resources. Furthermore, our framework admits 1-step solving process to obtain prompt adaptation, which is efficient in time and requires fewer resources than direct gradient descent training using aligned images and labels.**
> >
> > I agree that we may use partial alignment with some thresholds but tuning the threshold also takes time. Can you give some numbers about the sample alignment time on real-world dataset?

---

> > > ### Author Response · Authors · 2022-12-11
> > > **Alignment runtime estimation**
> > >
> > > Surely, this would be a nice suggestion.
> > >
> > > In a timely manner, the alignment time is computed with the datasets presented in our study, namely: [**DNS challenge 2020 (Speech Enhancement)**](https://www.microsoft.com/en-us/research/academic-program/deep-noise-suppression-challenge-interspeech-2020/) & [**CUFED images (Super Resolution)**](https://acsweb.ucsd.edu/~yuw176/event-curation.html). Both datasets are of the **real world** with details described in the Experimental Section and the Appendix.
> > >
> > > > **[Detailed implementations]** The Optimal Transport (OT) alignment is implemented on top of our original LVA experiments, following the construction of the paper [https://jmlr.org/papers/v22/20-451.html](https://jmlr.org/papers/v22/20-451.html), where the Wasserstein distance is the key ingredient. When given designated target samples, the alignment is performed batch-wisely over the source domain so that the batch size of the source domain also contributes to the runtime. Fixing the hardware, the OT alignment was executed **single-threadedly on one CPU**: Intel(R) Xeon(R) Gold 6152 CPU @ 2.10GHz. As many factors may contribute to the processing time, the following results are averages of **1000 runs** with the temporal unit: *second*.
> > >
> > > > **Table 1: SE alignment time, averaged over 1000 runs (unit:  second).**
> > > |  | DNS challenge |  | (SE experiment) |  |
> > > |:---------------|:------:|------:| :------:| :------:|
> > > | **target samples** |  **/ source batch size** | 100  | 1000 | 10000 |
> > > | 16 |  | 0.0041 | 0.0659 | 0.6094 |
> > > | 32 |  | 0.0030    | 0.0719 | 0.6112 |
> > > | 64 |  | 0.0032 | 0.1064 | 0.8358 |
> > > | 128 |   | 0.0045 | 0.1196 | 0.6576 |
> > >
> > >
> > > > **Table 2: SRCNN alignment time, averaged over 1000 runs (unit:  second).**
> > > |  | CUFED images |  | (SRCNN experiment) |  |
> > > |:---------------|:------:|------:| :------:| :------:|
> > > | **target samples** |  **/ source batch size** | 100  | 1000 | 10000 |
> > > | 16 |  | 0.0005 | 0.0011 | 0.0514 |
> > > | 32 |  | 0.0007 | 0.0013 | 0.1229 |
> > > | 64 |  | 0.0010 | 0.0016 | 0.0965 |
> > > | 128 |   | 0.0010 | 0.0072 | 0.0512 |
> > >
> > > As a result, **Table 1** shows that aligning the entire source domain of the DNS challenge (totaling 112,000 samples) to the target samples takes at most **13.3952 sec**, which would not be a dominant portion with respect to the training time. Particularly, this is only the result of a single thread with 1 CPU. In multi-thread processing and full parallelization, the alignment time is expected to significantly decrease.
> > >
> > > If the reviewer considers the above information helpful, it can be amended as supplementary material to a revision whenever possible to inform interested readers.

---

> > > > ### Comment · Reviewer_xrcZ · 2022-12-12
> > > > **Thanks for your response**
> > > >
> > > > I have read the training time and I would encourage the authors to release the code and include some time analysis in the paper. I increased my score to 6.

---

> > > > > ### Author Response · Authors · 2022-12-12
> > > > > **Authors thank the reviewer for the constructive comments**
> > > > >
> > > > > The authors thank the reviewer for the constructive comments and recognizing the effort of this study.
> > > > >
> > > > > The time analysis and the associated code will be organized and released onto the same Github given in the manuscript, along with the existing codes of all experiments. The additional domain adaptation experiments conducted during the discussion period including **Office-31** & **MNIST $\to$ USPS** will also be amended accordingly whenever a next revision is available.

---

### Official Review · Reviewer_hjUS · 2022-10-25

**Confidence:** 4
**Correctness:** 3
**Technical Novelty And Significance:** 2
**Empirical Novelty And Significance:** 3
**Recommendation:** 5

**Clarity, Quality, Novelty And Reproducibility:**

**Clarity:**

The paper is mostly well-written with minor issues:

1. In definition 3.2, the authors use g for the upper layers. However, in equation 3.2, they changed to f for both pre-trained and upper layers. I don't understand why this change is necessary.
2. The format of citations sometimes has issues. For example in paragraph 2 in the introduction, these citations should be in parathesis (using \citep)
3. The use of mathematical notation is sometimes inappropriate. For example, $q_i \cong 0$ on page 5. The notation $\cong$ normally means "is congruent to". I am not sure why it is used here.

**Quality:**

I think there are probably no mathematical mistakes in their theoretical analysis and their empirical study is also sufficient. However, I am not fully convinced by the following:
1. The definition of data deviation. According to their definition, if there is an extreme outlier in the new dataset, the data deviation can be very large even though other data points are exactly the same as those in the pre-training dataset. Because of that, it does not seem to be a good measure of data distance to me.
2. At the beginning of section 3.2, it is stated that neural networks are Lipschitz continuous. Do you mean all neural networks are Lipschitz continuous? It does not seem to be right to me.
3. Remark 3.7, only from equation 10, how can you tell that $\mathcal{L}(g)$ will become large if $\epsilon_{\text{data}}$ is large? Equation 10 is inequality so $\mathcal{L}(g)$ can be very small even if the bound on the right is large.

In addition to the above, I am also concerned by the high-level issues mentioned in the weaknesses section.

**Novelty:**

It may be the first time a similar analysis and the proposed method are used for transfer learning. However, the idea behind the proposed method is not entirely new (see the weaknesses section above), and their theoretical analysis does not capture the important aspects of transfer learning.

**Reproducibility:**

The code is provided and the experimental details are sufficient.

**Details Of Ethics Concerns:**

I have no ethical concern

**Strength And Weaknesses:**

**Strength:**

1. Most technical analysis is correct and their empirical study is also sufficient.
2. The proposed approach which directly solves the linear functional on top of the pre-trained feature extraction show promising results on downstream tasks

**Weaknesses:**

1. Their analysis does not capture many important aspects of network-based transfer learning. Firstly, they only discuss the training loss but do not consider the generalization ability of models on test data. But achieving good training loss does not necessarily mean the learning is successful. Secondly, SGD optimization is normally used to train neural networks, their analysis does not consider the complexity of optimization. And their variational analysis in function space does not capture the optimization procedure in parameter space.
2. While the proposed method based on their analysis is interesting in the transfer learning setting, the idea is not entirely new. Similar ideas already exist in the meta-learning literature. ([Meta-Learning with Differentiable Closed-Form Solvers](https://arxiv.org/abs/1805.08136), [Meta-Learning with Differentiable Convex Optimization](https://openaccess.thecvf.com/content_CVPR_2019/papers/Lee_Meta-Learning_With_Differentiable_Convex_Optimization_CVPR_2019_paper.pdf), [Meta-Learning Priors for Efficient Online Bayesian Regression](https://arxiv.org/abs/1807.08912)). I am not sure if it is the first time the idea is applied to transfer learning. Moreover, although I can see the connection between the proposed method and their theoretical analysis, I don't think the theoretical part is absolutely necessary to come to their proposed approach. They could simply talk about their method without the layer variational analysis.

**Summary Of The Paper:**

This paper provides a theoretical analysis of network-based transfer learning using layer variational analysis. Their analysis is claimed to prove that the success of transfer learning is guaranteed with certain data conditions. Based on their analysis, they also propose an alternative method for network-based transfer learning, which demonstrates better efficiency and accuracy for domain adaptation.

**Summary Of The Review:**

I recommend rejection of this paper because of the following:
Their theoretical analysis does not capture several important aspects of transfer learning (see Weaknesses section) and their analysis has a few potential technical issues (see Quality section). The proposed method based on their analysis may not be seen as entirely new to me because similar ideas do exist in meta-learning literature (see Weaknesses section).

---

> ### Author Response · Authors · 2022-11-20
> **Response to Reviewer hjUS**
>
> > **[Clarity]  The paper is mostly well-written with minor issues: In definition 3.2, the authors use $g$ for the upper layers. However, in equation 3.2, they changed to $f$ for both pre-trained and upper layers. I don't understand why this change is necessary.**
>
> In our study, the notation $f = f_n \circ \cdots \circ f_1$ (of layers $f_j$) is used denote the pretrained network and $g = (g_n \circ \cdots\circ g_{k+1}) \circ (f_k \cdots \circ f_1)$ as the finetuned network, where the first $k$ layers $f_1, \ldots, f_k$ **are fixed** and the rest $g_{k+1}, \ldots, g_n$ layers **are finetuned**. We have re-examined the manuscript to make sure the notations are consistent.
>
> > **[Clarity] The format of citations sometimes has issues. For example in paragraph 2 in the introduction, these citations should be in parathesis (using "citep")**
>
> We are grateful for the reviewer's feedback, and the citations have been changed to **"citep"** in accordance with the reviewer's suggestion.
>
> > **[Clarity] The use of mathematical notation is sometimes inappropriate. For example,  on page 5. The notation  normally means "is congruent to". I am not sure why it is used here.**
>
> We thank the reviewer for pointing this out. Indeed, the notation $\cong$ is often used in the group theory as congruence. In the previous manuscript, we intended to mean "*approximately equal to*" and now we have switched to the symbol "$\approx$" accordingly in the revised manuscript.
>
>
> > **[Quality] I think there are probably no mathematical mistakes in their theoretical analysis and their empirical study is also sufficient. However, I am not fully convinced by the following: The definition of data deviation. According to their definition, if there is an extreme outlier in the new dataset, the data deviation can be very large even though other data points are exactly the same as those in the pre-training dataset. Because of that, it does not seem to be a good measure of data distance to me.**
>
> Indeed, the definition is restricted for clear statements of our analysis, written under the intention for readers to easily see and capture the essential ideas. The definition certainly can be extended to a more general form containing outliers in a new dataset, while a delicate description will then be set up to exclude them from our network computing/learning.
>
> > **[Quality] At the beginning of section 3.2, it is stated that neural networks are Lipschitz continuous. Do you mean all neural networks are Lipschitz continuous? It does not seem to be right to me.**
>
> In this work, we consider neural networks of the form Eq. (1) [in Definition 3.1]. Therefore, under the compositions of (Lipschitz) activation functions and affine layers, the neural network of Eq. (1) is necessarily Lipschitz continuous. The statement is based on the following fact that for two Lipschitz functions $h_1: X \to Y$, $h_2: Y \to Z$, the composition $h_2 \circ h_1: X \to Z$ is again Lipschitz continuous:
> $$ || h_2 \circ h_1 (x) - h_2 \circ h_1 (y) || \leq  C_{h_2} || h_1(x) - h_1(y) || \leq  C_{h_2} C_{h_1} || x - y || $$
> Therefore, as long as the chosen activation functions $\sigma_j$ are all Lipschitz continuous, the corresponding neural network of the form Eq.(1) stays Lipschitz continuous, which is the case for most activation functions, including ReLu, Tanh, Logistic, Sigmoid, and Softmax, etc.
> We have revised the statement in Sec. 3.2 to emphasize the form of Eq. (1) and the Lipschitz activations.
>
> > **[Quality] Remark 3.7, only from equation 10, how can you tell that $\mathcal{L}(g)$ will become large if $\epsilon\_{\text{data}}$ is large? Equation 10 is inequality so $\mathcal{L}(g)$ can be very small even if the bound on the right is large.**
>
> We apologize for the confusion caused by the inaccurate statement. Indeed, we only know the upper bound of $\mathcal{L}(g)$ becomes large if $\epsilon\_{\text{data}}$ is large. Our intention was to point out that the behavior of the upper bound fits what we expected when considering two very different datasets. Remark 3.7 has been revised accordingly in the manuscript.
>
> > **[Summary Of The Review] I recommend rejection of this paper because of the following: Their theoretical analysis does not capture several important aspects of transfer learning (see Weaknesses section) and their analysis has a few potential technical issues (see Quality section). The proposed method based on their analysis may not be seen as entirely new to me because similar ideas do exist in meta-learning literature (see Weaknesses section).**
>
> We appreciate the reviewer's helpful comments and suggestions. We provide point-by-point responses for clarification, which have been incorporated into our revision.

---

> ### Author Response · Authors · 2022-11-20
> **Response to Reviewer hjUS**
>
> >  **2. While the proposed method based on their analysis is interesting in the transfer learning setting, the idea is not entirely new. Similar ideas already exist in the meta-learning literature. ([Meta-Learning with Differentiable Closed-Form Solvers](https://arxiv.org/abs/1805.08136), [Meta-Learning with Differentiable Convex Optimization](https://openaccess.thecvf.com/content_CVPR_2019/papers/Lee_Meta-Learning_With_Differentiable_Convex_Optimization_CVPR_2019_paper.pdf), [Meta-Learning Priors for Efficient Online Bayesian Regression](https://arxiv.org/abs/1807.08912)). I am not sure if it is the first time the idea is applied to transfer learning. Moreover, although I can see the connection between the proposed method and their theoretical analysis, I don't think the theoretical part is absolutely necessary to come to their proposed approach. They could simply talk about their method without the layer variational analysis.**
>
> 3. The paper [*Meta-Learning with Differentiable Closed-Form Solvers*](https://arxiv.org/abs/1805.08136) shares a similar idea as the study *Meta-Learning with Differentiable Convex Optimization*, where the authors began with a general assumption in meta-learning where a common latent embedding $\phi$ is used to map to the same feature space. Therefore, as stated above, this is one major difference to set apart with our study.
>
>     Furthermore, their main proposal is to invoke a linear episodic predictor $f(\phi(x)) = \phi(x) \cdot W$ for final classification using the pseudo-inverse: $$ W = (\phi(X)^T \phi(X) + \lambda I )^{-1} \, \phi(X)^T \, Y $$
>
>     Again, by comparing this equation to our Eq. (21): $$ \delta f_n =\left( \widetilde{z}^T\cdot \widetilde{z} \right)^{-1} \widetilde{z}^T \cdot q $$
>     We immediately recognize the difference: their regression targets are simply labels $Y$; ours is a non-trivial combination $q_i = \delta y_i - J(f)(x_i) \cdot \delta x_i + \left( f(x_i) - y_i \right)$ derived from LVA, which tells us the memory of a source domain is carried over to a target domain. This becomes a signature allowing us to easily see the difference.
>
> In summary, despite that the listed studies being interesting and inspiring from their own perspectives, their major considerations do not seem to coincide with ours in many aspects. By addressing closely to the differences, we hope the reviewer's concern can be cleared.

---

> ### Author Response · Authors · 2022-11-20
> **Response to Reviewer hjUS**
>
> >  **2. While the proposed method based on their analysis is interesting in the transfer learning setting, the idea is not entirely new. Similar ideas already exist in the meta-learning literature. ([Meta-Learning with Differentiable Closed-Form Solvers](https://arxiv.org/abs/1805.08136), [Meta-Learning with Differentiable Convex Optimization](https://openaccess.thecvf.com/content_CVPR_2019/papers/Lee_Meta-Learning_With_Differentiable_Convex_Optimization_CVPR_2019_paper.pdf), [Meta-Learning Priors for Efficient Online Bayesian Regression](https://arxiv.org/abs/1807.08912)). I am not sure if it is the first time the idea is applied to transfer learning. Moreover, although I can see the connection between the proposed method and their theoretical analysis, I don't think the theoretical part is absolutely necessary to come to their proposed approach. They could simply talk about their method without the layer variational analysis.**
>
> We appreciate the reviewer for sharing other interesting literature on meta-learning. After close examination, it can be found that our approach does **not** share fundamental ideas and considerations with the papers from our perspective. The reasons are as follows:
>
> 1. In the paper: [*Meta-Learning Priors for Efficient Online Bayesian Regression*](https://arxiv.org/abs/1807.08912), the **starting point** was to find the posterior probability density $q_{\xi}(y | x, D_t)$ of parameters $\xi$ close to the ground truth probability $p(y | x, D_t)$ using KL-divergence in (their) Eq. (3): $$ \min D_{KL}( p(y | x, D_t) || q_{\xi}(y | x, D_t)) $$
>
>     The authors then considered an **alternative accessible problem** of tuning the sample distribution $D^*_{\tau} = \{ (x_t, y_t) \}_{t=1}^{\tau}$ using **Bayesian function regression**. Consequently, the authors in (their) Section 3 assumed the data $y_t$ resulting from the form $y_t = \phi^T(x_t) K + \epsilon_t$ with optimal functions $\phi$ to be found.
>
>     Their considerations were mainly **varying the probability distribution** to fit the observed data, whereas our construction is to **directly vary the network** by weights $(W_n$, $b_n) \Leftrightarrow f_n$ for finetuning. In this view, this paper appears to be more close to the Generative Adversarial Net (GAN) and the Variational Auto-Encoder (VAE) type of approach, which sets out from data probability approximation. Moreover, our construction does not really concern probabilities.
>
> 2. In the paper: [*Meta-Learning with Differentiable Convex Optimization*](https://openaccess.thecvf.com/content_CVPR_2019/papers/Lee_Meta-Learning_With_Differentiable_Convex_Optimization_CVPR_2019_paper.pdf), the authors assumed the base learner $\mathcal{A}$ in a convex form. Specifically, they considered a multi-class SVM $\theta = (w_k )$ and optimized $\(w_k )$ to classify training data $D^{\text{train}} = \{(x_n, y_n)\}$ from a latent embedding $f_{\phi}(x_n)$ (see Eq.(4)).
>
>     There are **two major distinctions** between our essential ideas. First, we **do not** assume the training data and the test data resulted from the same latent (feature) embedding $f_{\phi}$. Since meta-learning in general intends to approach such $f_{\phi}$ or $\phi$, the study may eventually rely on a simple SVM classifier $\theta =  (w_k)$ and $\phi$ to achieve meta-learning by (their) Eq.(4), (12). In fact, our study specifically begins from domains with deviation (computed by our Definition 3.4), and therefore we emphasize how the *network overcomes the data deviation by properly adjusting the network weights using variational analysis*. This eventually leads to our **core concept** Eq. (17) \& (20), where the **transferal residue** indicates the pretrained error can be accumulated to the new target domain whenever a network is subject to finetuning. This key equation, our Eq.(20), cannot find its counterpart in their study.
>
>     Second, one of our objectives is to describe the finetune bound and subsequently to know the worst case, where this does not seem to be in their concerns.

---

> ### Author Response · Authors · 2022-11-20
> **Response to Reviewer hjUS**
>
> >  **1. Their analysis does not capture many important aspects of network-based transfer learning. Firstly, they only discuss the training loss but do not consider the generalization ability of models on test data. But achieving good training loss does not necessarily mean the learning is successful. Secondly, SGD optimization is normally used to train neural networks, their analysis does not consider the complexity of optimization. And their variational analysis in function space does not capture the optimization procedure in parameter space.**
>
> We deeply appreciate the valuable comments. Indeed, our study does not directly discuss the generalization ability of models on test sets. This is mainly because we intended to simply our framework so that the learning bounds can be tangible and the subsequent finetune computation can be carried out analytically. In fact, to clearly describe the finetuning mechanism, certain efforts are required, and it would quickly become cumbersome if to include extra conditions into consideration. It is certainly possible to extend our current setting into a framework with test set generalization. From this perspective, we appreciate the reviewer for reminding us to explore this direction.
>
> It is true that training loss does not necessarily mean the learning is successful. With the same consideration as the reviewer's, we conducted experiments on real-world applications and large datasets to see that our proposal is not far from reality. In fact, in measuring the performance of speech enhancement and image de-blurring, the perceptual measurements (PESQ, STOI, PSNR, etc) were used to conform with the standards of the corresponding field for fair comparisons. Various tasks and experiments have successfully verified that the LVA is a method with generalization ability. Moreover, the concern of overfitting can be mitigated by regularizations, which are commonly seen.
>
> Due to the nature of our LVA method, the time-consuming optimization process by gradient computations can be replaced so that the loss may soon reach the minimum. This serves as an advantage for us to promptly derive a transferred network in certain cases. Although there are exchanges in space complexity and time complexity in trading from SGD to LVA, our approach yields an accurate finetuning net and provides a fresh perspective for transfer learning.

---

> > ### Comment · Reviewer_hjUS · 2022-11-23
> > **Response to Authors**
> >
> > Thanks for the authors' response. After reading the comments, I still think generalization is a fundamental element in this analysis. Without a measure of generalization, it may not be convincing to talk about any kind of learning (including transfer learning) at all. And I think empirical results cannot replace theoretical analysis in this case. Thanks for providing a detailed explanation of how the proposed method is different from the references I provided. It might be good to include some of them in the related work section when appropriate. The authors have also fixed many of their technical issues. In general, I believe this work still needs further development and I will keep my original score.

---

> > > ### Author Response · Authors · 2022-11-28
> > > **Response to Reviewer's concern on generalizations**
> > >
> > > We thank the reviewer for recognizing the unique contribution of our work in distinguishing it from the mentioned references. The related works in meta-learning are indeed inspiring and complement our investigation from different perspectives. Therefore, if a revision is allowed, the related references will certainly be amended to our study.
> > >
> > > On the other hand, as mentioned in our previous response, the generalization bound of our proposal can be **estimated under the presence of a test set** $\widetilde{\mathcal{D}}_{\text{test}}$ in addition to the adaptation set $\widetilde{\mathcal{D}}$, where we originally intended to present a *succinct* framework for readers to grasp easily. Following the setting and **Theorem 3.6** of our study, the generalization ability is as follows:
> > >
> > >
> > > > **Theorem (Generalization error bound)**
> > > Given a test (held-out) set $\widetilde{\mathcal{D}}_{\text{test}} = \{ (\widetilde{x}^{\text{(test)} }\_i , \widetilde{y}^{\text{(test)}}\_i ) \}\_{i=1}^{N\_{test}}$ with $| \widetilde{\mathcal{D}}\_{test} | \leq | \widetilde{\mathcal{D}} |$, a **well-finetuned** net $g$ of Theorem 3.6 has a generalization error bound,
> > > $$ \mathcal{L}\_{test}(g) = \frac{1}{N\_{test}} \sum\_{i=1}^{N\_{test}}  || g(\widetilde{x}^{(test)}\_i) - \widetilde{y}^{(test)}\_i ||\_{\mathcal{Y}}^2  \leq C \varepsilon^2\_g +  3 \mathcal{L}(g) $$ where $\varepsilon\_g$ is the data deviation between $\widetilde{\mathcal{D}}$ and $\widetilde{\mathcal{D}}\_{\text{test}} $ computed by Definition 3.4; $C$ is a constant depending on the Lipschitz constant $C\_g$, and $ \mathcal{L}(g)$ is the finetune loss from Theorem 3.6.
> > >
> > > This result indicates that the generalization error $\mathcal{L}\_{test}(g) $ is jointly bounded by the **(training) finetuned loss** $\mathcal{L}(g)$ as well as the **intrinsic deviation** between the held-out set and the training set $\varepsilon\_g$. Consequently, when the adaptation finetuning is well conducted and the test set is not singular from the training set, the generalization error is naturally contained, and thus the generalization ability follows.
> > >
> > > Moreover, to consider more *generalized measurements other than norms* on the output space $\mathcal{Y}$, we can extend our theory onto a [**metric space**](https://en.wikipedia.org/wiki/Metric_space) $(\mathcal{Y}, d\_{\mathcal{Y}})$, one of topological spaces with distance notion $d\_{\mathcal{Y}}: \mathcal{Y} \times \mathcal{Y} \to \mathbb{R}$ satisfying $\forall y, y\_1, y\_2, y\_3 \in \mathcal{Y}$:
> > > $$ (1) \quad  d\_{\mathcal{Y}} (y, y) = 0 $$
> > > $$ (2) \quad d\_{\mathcal{Y}} (y\_1, y\_2) > 0 , \quad \text{if $y\_1 \neq y\_2$} $$
> > > $$ (3)  \quad d\_{\mathcal{Y}} (y\_1, y\_2) = d\_{\mathcal{Y}} (y\_2, y\_1) $$
> > > $$ (4)  \quad d\_{\mathcal{Y}} (y\_1, y\_2) \leq d\_{\mathcal{Y}} (y\_1, y\_3) + d\_{\mathcal{Y}} (y\_3, y\_2) $$
> > >
> > > As such, a generalized measurement can be defined using the **metric** $d\_{\mathcal{Y}}$:
> > > $$  \mathcal{L}\_{\text{generalized test}}(g) = \frac{1}{N\_{test}} \sum\_{i=1}^{N\_{test}}  d\_{\mathcal{Y}} (\widetilde{y}^{(predict)}\_i , \widetilde{y}^{(label)}\_i ) $$
> > >
> > > and the associated **(accuracy) score function** $s: \mathcal{Y} \times \mathcal{Y} \to \mathbb{R}$ can be defined inversely, e.g., by $ s (\widetilde{y}^{(predict)} , \widetilde{y}^{(label)} )   = c - d\_{\mathcal{Y}} (\widetilde{y}^{(predict)}, \widetilde{y}^{(label)})$ or $ s (\widetilde{y}^{(predict)}, \widetilde{y}^{(label)} )   = c / d\_{\mathcal{Y}} (\widetilde{y}^{(predict)} , \widetilde{y}^{(label)})$ for some $c \in \mathbb{R}$ so that the higher score indicates the better learning performance. A general metric $d\_{\mathcal{Y}}$ will suffice to *represent a wide class* of perceptual measurements, such as empirical scores: *PESQ, STOI, PSNR, etc* as well as some divergences: e.g. [Jensen–Shannon divergence](https://en.wikipedia.org/wiki/Jensen%E2%80%93Shannon_divergence). With the definition, one can compute the generalization performance on a held-out set.
> > >
> > > However, **without knowing in advance what (abstract) score is to be encountered**, a general bound is therefore considered: $d\_{\mathcal{Y}} (y\_1, y\_2) \leq K || y\_1 - y\_2 ||\_{L\_2}$, with a constant $K$ bridging the abstract measurement and the Euclidean norm. Consequently, the generalized measurement on test set derives a similar bounded loss:
> > > $$  \mathcal{L}\_{\text{generalized test}}(g) = \frac{1}{N\_{test}} \sum\_{i=1}^{N\_{test}}  d\_{\mathcal{Y}} ( g(\widetilde{x}^{(test)}\_i) , \widetilde{y}^{(test)}\_i ) \leq K (C \varepsilon^2\_g +  3 \mathcal{L}(g) )$$
> > > up to a constant $K$ to be determined in numerical experiments.
> > >
> > > We believe that this analysis should address the reviewer's generalization concern. The above computations and detailed proof can be amended to a next revision if the reviewer considers them helpful for readers to understand our complete development.

---

> > > > ### Comment · Reviewer_hjUS · 2022-12-06
> > > > **Response to Authors**
> > > >
> > > > Thank you for the great efforts to derive the generalization error bound. If I understand correctly, the intrinsic deviation could be very large if there is an extreme outlier in the test set, in which case the bound is not very useful. In my opinion, the authors should reconsider the definition of data deviation. Maybe the authors could adopt a probabilistic perspective, and treat data as samples from the data distribution.
> > > >
> > > > As a final remark, I am still not convinced that the analysis presented in this paper could provide us with significant insight. This paper makes lots of assumptions during their analysis, and some of them may not capture the essence of network-based transfer learning. As a personal opinion, it might even be better if this paper simply focuses on their proposed LVA-inspired methodology and cuts short their analysis.

---

> > > > > ### Author Response · Authors · 2022-12-10
> > > > > **Re: regarding the restrictions and assumptions**
> > > > >
> > > > > We thank the reviewer for recognizing the generalization error bound derived so that we have completed the initial request of **demonstrating generalization error bound exists for our method.**
> > > > >
> > > > > On the other hand, the reviewer is correct that if an *extreme* outlier exists and the intrinsic deviation may increase to give a loose bound. However, facing the same situation we believe most learning algorithms will equally fail because **intuitively it would not be possible for an algorithm to learn from dogs-and-cats images, but asked to recognize an orange in a test set**. Indeed, considering the [**Vapnik–Chervonenkis dimension**](https://en.wikipedia.org/wiki/Vapnik%E2%80%93Chervonenkis_dimension), one already knows that certain extreme outlines can result in difficulty in *shattering a dataset* for a learning algorithm, let alone guarantee well generalization on a testing set. In this view, extreme outliers are generally a prevailing problem, not merely pertaining to our method. Naturally, our mathematics **faithfully tells us** that the generalization error bound has to be loose in such a case.
> > > > >
> > > > > Regarding the assumptions and restrictions, it is our acknowledgment that unavoidably there are within our theoretical method. However, in the fundamental [**Probably Approximately Correct (PAC) learning**](https://en.wikipedia.org/wiki/Probably_approximately_correct_learning) and the [**Vapnik–Chervonenkis (VC) theory**](https://en.wikipedia.org/wiki/Vapnik%E2%80%93Chervonenkis_theory), one recalls that the original theories **only considered binary classifiers**. According to today's standards, this can be extremely restrictive and outdated. However, they are still deemed as the current foundations of Statistical Learning and one would not *deny the inspirations* that the theories with limited assumptions have brought us. It is due to certain assumptions that the concrete properties can be manifest.
> > > > >
> > > > > This work not only aims to propose a novel method but is also bound to interpret the transfer learning mechanism. Most existing studies focus on experimentation and empirical discussions, while many theories were built on high ground, sometimes too far from reality. It is our attempt to provide a reasonable interpretation for the behaviors of transfer learning and identify a balance where exact calculations can be derived as well as applications can be made. Bearing this goal in mind, our setting has its inherent limitation. In return, we are able to pin down the concept of **knowledge transfer** to precise mathematical terms; we believe that this contribution is unique. Numerical experiments further confirm our novel perspective is beyond pure imagination. It is our humble hope that this study can be valued from both theoretical and application perspectives.

---

### Author Response · Authors · 2022-12-12
**Concluding Remarks**

The authors sincerely thank all Reviewers for their constructive comments and valuable time. Reaching the end of the discussion period, we like to briefly summarize our feedback during the discussion:

> **To Reviewer hjUS**’s concern that our method is **similar to existing literature**, the suggested references were looked into to find out and clarify why the settings are distinct from our approach in many aspects. Detailed comparisons are described in our response. Another question regarding **only considering the training loss**, two mathematical proofs for our generalization (test) loss bounds are provided in the discussion to demonstrate the generalization ability exists and can be further extended to non-trivial (perceptual) losses.

> **To Reviewer xrcZ**'s interest in seeing more experiments. Additional adaptation tasks using image datasets **Office-31** & **MNIST $\to$ USPS** were conducted to further demonstrate the *real-world applicability* of our proposed method. Detailed quantitative analyses described in the response are also amended in the Appendix of the revision. Another question regarding **sample alignment**, time analysis is subsequently provided to show that the alignment does not occupy the major computational expense.

> **To Reviewer 2KJc**'s close examination of various technical parts, with deep appreciation the authors extended their presentation and formulation in the revision to hopefully release similar concerns from interested readers. The suggestions to improve our SE experiment, a pretrained **FullSubNet** is subsequently implemented to serve as a new baseline model. Detailed analysis and corresponding results are presented in the discussion.


In light of all the helpful suggestions and comments, the manuscript is further improved, while the original intention remains:

This work utilizes a novel mathematical technique **Layer Variational Analysis (LVA)** to characterize the transfer learning mechanism. Most existing studies focus on experimentation and empirical discussions, while many theories were built on high ground, sometimes too far from reality. Our attempt is the **first to provide intuitive interpretations** for the transfer learning behaviour and **to construct a theory** where mathematical calculations can be derived as well as practical implementations can be made.

Aiming towards this goal, the LVA successfully analyzes transfer learning as well as provides an optimal solution on a theoretical basis. Consequently, this optimal solution allows us to pin down the concept of **knowledge transfer to precise mathematical terms**. To our best knowledge, *the contribution is fundamental and unique*. Numerical experiments further confirm our novel perspective is beyond pure imagination. It is our humble hope that this study can be valued from both *theoretical and applied* perspectives, which eventually result in a profound influence on the machine learning society.

---

### Decision · Program_Chairs · 2023-01-20

**Decision:**

Accept: poster

**Justification For Why Not Higher Score:**

The are few issues with the proposed technique that have been  addressed only partially.

**Justification For Why Not Lower Score:**

The paper discuss a challenging problem, and the proposed idea is interesting and sound. The paper is well written. The technical novelty is sufficient.

**Metareview: Summary, Strengths And Weaknesses:**


The authors aim to tackle a very challenging issue, namely  establishing both formal derivations and heuristic analysis to formulate the theory of transfer learning in deep learning. To that end, the authors put forth a framework based on  layer variational analysis and show that  that the success of transfer learning can be guaranteed with corresponding data conditions. The paper conducts experiments on a synthetic dataset, a speech enhancement task, and image de-blurring/super-resolution.

The paper is nicely organized and well-written, the code is provided and the experimental details are sufficient. The major novel aspect is the method of layer-wise optimization, and  Eqs. (17)-(18) give new insights to transfer learning. In the first review round, the reviewers however rises several  concerns, specifically about some technical aspects of the proposed solution, and the experimental validation. The authors did a great job in responding to the reviewers’ comments, and most of the negative comments have been properly addressed and solved. One of the reviewers has also increased his/her initial score. A few comments are still pending, i.e., the authors have not provided a determined  response to the following issues: (i) Generalization ability of the model on the test data. Nonetheless, the authors have given a generalization error bound, which may partially address the reviewer's concern. (ii) The framework requires sample alignment, which can have a  huge cost and may not be applicable in most fine-tuning cases. Nonetheless, the authors have provided a reasonable was to (practically) address the problem, and  (iii) A few of the assumptions made by the authors may not always capture the essence of the network-based transfer learning

 In sum, the paper is nicely written with some elements of novelty, and it also  provides a good deal of experimental results and analyses to back-up the authors’ claim. A few concerns remain, but the paper could be considered a first step to establish a theoretical analysis of network-based transfer learning.

**Note From Pc:**

if the above contains the word "oral" or "spotlight" please see: "oral" presentation means -> notable-top-5% and "spotlight" means -> notable-top-25%. As stated in our emails, we are disassociating presentation type from AC recommendations

**Summary Of Ac-Reviewer Meeting:**

I have invited all of the reviewers to have a virtual meeting, but none of the reviewers has replied to my doodle invitation. Nonetheless, all reviewers have been responsive whenever I have asked them to read and ack the authors' responses.